# Divide, Conquer, and Coalesce: Meta Parallel Graph Neural Network for IoT Intrusion Detection at Scale

## ABSTRACT

This paper proposes Meta Parallel Graph Neural Network (MPGNN) to establish a scalable Network Intrusion Detection System (NIDS) for large-scale Internet of Things (IoT) networks. MPGNN leverages a meta-learning framework to optimize the parallelism of GNN-based NIDS. The core of MPGNN is a coalition formation policy that generates meta-knowledge for partitioning a massive graph into multiple coalitions/subgraphs in a way that maximizes the performance and efficiency of parallel coalitional NIDSs. We propose an offline reinforcement learning algorithm, called Graph-Embedded Adversarially Trained Actor-Critic (G-ATAC), to learn a coalition formation policy that jointly optimizes intrusion detection accuracy, communication overheads, and computational complexities of coalitional NIDSs. In particular, G-ATAC learns to capture the temporal dependencies of network states and coalition formation decisions over offline data, eliminating the need for expensive online interactions with large IoT networks. Given generated coalitions, MPGNN employs E-GraphSAGE to establish coalitional NIDSs which then collaborate via ensemble prediction to accomplish intrusion detection for the entire network. We evaluate MPGNN on two real-world datasets. The experimental results demonstrate the superiority of our method with substantial improvements in F1 score, surpassing the state-of-the-art methods by 0.38 and 0.29 for the respective datasets. Compared to the centralized NIDS, MPGNN reduces the training time of NIDS by 41.63% and 22.11%, while maintaining an intrusion detection performance comparable to centralized NIDS.

## KEYWORDS

Network intrusion detection, graph neural network, offline reinforcement learning, scalability

## 1 INTRODUCTION

The Internet of Things (IoT) [11] has brought about a paradigm shift in the way we interact with the environment, enabling seamless connectivity and automation across diverse domains such as smart homes, healthcare, and transportation. According to the forecast from the International Data Corporation, the number of IoT devices connected to the Internet is projected to surpass 41 billion by 2025 [22]. The massive connectivity of IoT networks is increasingly challenged by nonconventional security risks. Millions of IoT devices are collecting a vast trove of sensitive information, encompassing location data, health records, biometric features, and user behavioral patterns, which, if not adequately safeguarded, may become susceptible to severe privacy breaches and potential misuse by malicious entities [1]. Furthermore, the inherent resource limitations of IoT devices only allow minimal deployments of defensive strategies onboard, exacerbating the vulnerability of IoT devices to security risks. Consequently, there is a pressing need for a security solution to enhance the overall security posture of large-scale IoT networks and safeguard the privacy and integrity of IoT data.

The Network Intrusion Detection System (NIDS) serves as a robust defense mechanism against potential threats in IoT networks. By actively monitoring suspicious activities and unauthorized access attempts, NIDS can promptly identify malicious behaviors and implement mitigation measures. Earlier works on IoT NIDS primarily relied on rule-based and signature-based techniques [5, 29], which often encounter limitations when confronted with previously unseen attacks. Therefore, IoT NIDS gradually incorporates machine learning (ML) methods, deep learning (DL) in particular, to enable the detection of complex and evolving threats by directly learning from network flow data and identifying patterns that may not be captured by rule-/signature-based approaches. Among state-of-the-art DL techniques for NIDS, Graph Neural Networks (GNNs) have emerged as the focal point of attention [10, 17, 35, 37]. NIDS typically operates on network flow data (e.g., NetFlow [7]) which can naturally be represented in a graph format. The detection of malicious lows relies on extracting insights from both the topological information and the details embedded in edge features. GNNs are tailored for processing graph-structured data, and are able to leverage the inherent structure of the graph by incorporating relational inductive biases into the DL architecture. This capability empowers GNNS to efficiently learn, reason, and generalize from IoT network traffic data, enabling GNN-based NIDS to capture complex relationships and dependencies among network entities and identify malicious behaviors.

While GNN-based NIDSs have demonstrated satisfactory performance on open datasets [10, 17, 35, 37], they have overlooked several essential factors that are critical for the practical implementation of NIDS in real-world IoT networks. **1)** The rapid expansion of IoT networks makes it thorny to ensure the timeliness of GNN-based NIDS. To accomplish network intrusion detection with GNNs, NIDS must collect IoT network states to construct a graph to represent device status (i.e., node features) and their interaction patterns (i.e., edge features). However, for large-scale IoT networks, the graph data size can reach the order of gigabytes, leading to substantial communication and latency overhead during the graph construction process. Moreover, the computational complexity of processing a massive graph exceeds the capabilities of resource-constrained IoT devices, leading to unacceptable detection delays in detecting intrusions. These challenges necessitate a scalable NIDS capable of handling the sheer volume of IoT devices and ensuring timely and responsive intrusion detection. **2)** Parallel and distributed processing is an intuitive solution to improve the scalability of GNN-based NIDS for large-scale IoT networks. However, the inherent properties of GNN pose significant challenges to achieving parallelism. In contrast to traditional DL techniques that typically operate on structured and independent training samples (e.g., image processing), GNNs inherently capture the dependencies among training samples through interconnections between vertices. Commonly used distributed learning frameworks (e.g., Federated Learning [19, 33]) disrupt the analysis of vertex

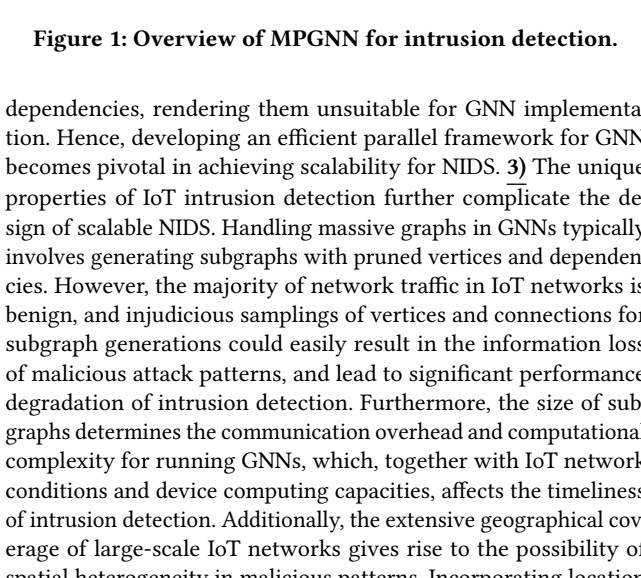

**Figure 1: Overview of MPGNN for intrusion detection.**

dependencies, rendering them unsuitable for GNN implementation. Hence, developing an efficient parallel framework for GNN becomes pivotal in achieving scalability for NIDS. **3)** The unique properties of IoT intrusion detection further complicate the design of scalable NIDS. Handling massive graphs in GNNs typically involves generating subgraphs with pruned vertices and dependencies. However, the majority of network traffic in IoT networks is benign, and injudicious samplings of vertices and connections for subgraph generations could easily result in the information loss of malicious attack patterns, and lead to significant performance degradation of intrusion detection. Furthermore, the size of subgraphs determines the communication overhead and computational complexity for running GNNs, which, together with IoT network conditions and device computing capacities, affects the timeliness of intrusion detection. Additionally, the extensive geographical coverage of large-scale IoT networks gives rise to the possibility of spatial heterogeneity in malicious patterns. Incorporating location awareness into IoT NIDS to account for this spatial heterogeneity holds great promise in enhancing intrusion detection performance.

To address the aforementioned challenges, we propose Meta Parallel Graph Neural Network (MPGNN) to establish a scalable NIDS for large-scale IoT networks. As illustrated in Fig.1, MPGNN facilitates the formation of regional coalitions and selects a high-end device (e.g., edge servers, smart home hubs, industrial gateways, etc.) as a coalition head to perform coalitional intrusion detection using GNNs. This enables the decomposition of a large-scale NIDS into multiple parallel coalitional NIDSs, which reduces the communication overhead and computational complexity of GNN-based intrusion detection to a scale that is manageable for coalition heads. Particularly, we develop a learning-based coalition formation policy for MPGNN, which is tailored to uncover the obscured impacts of dynamic network states, diverse device capabilities, and heterogeneous malicious patterns on the performance of coalitional NIDSs. The novelty and contribution of our method are summarized as follows:

1) MPGNN presents a novel meta-learning framework for parallel GNN-based NIDS in large-scale IoT networks. MPGNN consists of two loosely coupled phases: *coalition formation* and *coalitional intrusion detection*. The coalition formation policy generates meta-knowledge for partitioning the massive IoT network into multiple sub-networks (i.e., coalitions) in a way that maximizes the performance and efficiency of coalitional intrusion detection. In the phase of coalitional intrusion detection, E-GraphSAGE [17] is utilized to establish GNN-based NIDS for each coalition. E-GraphSAGE improves the representation of flow features and topological patterns, thereby boosting the capability of identifying malicious traffic flows. Coalitional NIDSs operate collaboratively via ensemble learning to complete the intrusion detection for the entire IoT network.

2) We propose an offline reinforcement learning method called Graph-Embedded Adversarially Trained Actor-Critic (G-ATAC) to learn a coalition formation policy. G-ATAC embeds Graph Convolutional Network (GCN) in the ATAC framework [6] to better characterize the impact of network topological information on the utility of coalition formation. G-ATAC optimizes coalition formation by jointly considering the intrusion detection accuracy, communication overheads, and computational complexities of coalitional NIDSs. In particular, G-ATAC captures the temporal dependencies of network states and coalition formation decisions over offline data, which avoids substantial communication overhead and latency incurred by interacting with large-scale IoT networks in an online manner.

3) We carry out systematic experiments on real-world datasets to evaluate the performance of MPGNN. Compared to the centralized NIDS, MPGNN achieves a 41.63% (22.11%) reduction in training time of GNN-based NIDS on the NF-ToN-IoT-V2 (NF-CSE-CIC-IDS2018-V2) dataset. MPGNN also reduces the scale of NIDS 3.3× while maintaining the performance of intrusion detection (in terms of accuracy, precision, recall, and F1-score) close to the centralized NIDS. Besides, in comparison with other parallel realizations, MPGNN improves F1 score by 0.38 and 0.29 on two datasets, respectively.

The rest of the paper is organized as follows: Section 2 reviews related works. Section 3 shows the design of MPGNN. Section 4 presents the experimental results, followed by the conclusion in Section 5. For readers of interest, pseudocodes of MPGNN, detailed experimental settings, and supplementary results are given in Appendix A, B, C, respectively.

## 2 RELATED WORKS

The advent of Deep Learning (DL) techniques, including Convolutional Neural Networks (CNNs) [16], Recurrent Neural Networks (RNNs) [34], have led to the proposal of DL-based NIDSs [14, 15, 36]. CNN-based NIDS [26, 28, 32] constructs network traffic as grid data. For instance, Xu et al. [32] described a flow as a grayscale image series, which can be fed into a 3D-CNN in chronological order to extract features reserving the relevance between packets. RNN-based NIDS [25, 36, 38] employ sequence data analysis for network flow identification. For example, Studiawan et al.[25] modeled the log messages in the system as sentences, and proposed a Gated Recurrent Unit (GRU) based sentiment analysis method to detect negative sentiment and consequently facilitate intrusion detection. However,

these methods overlook the spatial correlations of network traffic, hence limiting their ability to detect complex anomalies.

Recent approaches have endeavored to address this issue by leveraging graph-structured data to characterize the topological information of inter-node communications. Xiao et al. [30] proposed a graph embedding-based IDS, which utilized the first-order and the second-order graphs to learn the latent features from individual hosts and the global network, respectively. One notable limitation of [30] is its reliance on the transductive graph embedding method, which restricts its classification ability. Zhou et al. [35] employed a 12-layer Graph Convolutional Network (GCN) to capture the underlying structural information within the communication graph for botnet identification. However, this model focuses solely on extracting topological information related to botnets, neglecting the statistical features of network flows. Lo et al.[17] represented network flows as an attributed graph where the edge attributes correspond to the features of the network flows, and introduced a GNN-based edge classification model called E-GraphSAGE to realize intrusion detection. Zhu et al. [37] drew inspiration from graph theory and introduced the concept of line graph [13] to develop a Line-GraphSAGE algorithm for edge feature aggregation and network flow classification. Duan et al. [10] converted network traffic into a spatiotemporal line graph to incorporate both temporal and spatial information and utilized a dynamic line graph neural network (DLGNN) for intrusion detection. However, the previous research has primarily focused on improving the performance of intrusion detection, neglecting the evaluation of the suitability in field implementations.

The majority of precious works adopted the centralized learning paradigm, which is deemed unsuitable for large-scale IoT networks due to the significant communication overhead, high computational complexity, and large response time. Popoola et al. [19] proposed a Federated Deep Learning (FDL) based intrusion detection method, which adopted the DNN architecture to identify local anomalies while employing the Federated Averaging (FedAvg) algorithm to aggregate the updates of local models to overcome the absence of certain data categories in local networks. Yang et al.[33] proposed a cloud-edge collaboration-based intrusion detection method, which involved a Temporal Convolutional Network (TCN) at the edge and an FL architecture to coordinate multiple intrusion detection models across different parties. Although the above methods offer distributed realizations for NIDS, they may disrupt the analysis of network topological information using GNNs. In this paper, we propose a Meta Parallel Graph Neural Network (MPGNN) architecture, which achieves efficient parallelism for GNN-based NIDSs in large-scale IoT networks.

## 3 META PARALLEL GRAPH NEURAL NETWORK FOR PARALLEL NIDS

This section presents the framework of Meta Parallel Graph Neural Network (MPGNN) for parallel NIDS in large-scale IoT networks.

### 3.1 Parallel NIDS over Large-Scale IoT Networks

Our work considers typical scenarios of large-scale IoT networks, including smart cities [3], industrial Internets [23], and intelligent transportation systems[9]. These IoT networks comprise a diverse range of devices such as sensors, actuators, smart devices, and embedded systems, which are collectively indexed by $\mathcal{N} = \{1, 2, \ldots, N\}$. The computing capacity of these devices exhibits significant variations. For example, high-end devices (e.g., intelligent industrial gateways [24]) can be equipped with multi-core CPUs and general-purpose GPUs to support complicated computations, while low-end devices (e.g., sensors) only possess microcontrollers or low-power processors with minimal processing capabilities. As we will see later, this heterogeneity in device computing capacities offers an opportunity for establishing parallel GNN-based NIDS.

NIDS for IoT networks operates on flow-based network data. These flows are identified by flow endpoints (e.g., IP address, L4 port number, L4 protocol of IoT devices), and annotated by a set of flow fields (e.g., the number of packets, number of bytes, flow duration, etc.) that provide details about the flows. GNN-based NIDSs represent flow data in a graph format $G = \langle \mathcal{V}, \mathcal{E} \rangle$, where the flow endpoints are mapped to graph node set $\mathcal{V}$, and network flows are mapped to graph edge set $\mathcal{E}$. The GNN takes graph $G$ as the input and analyzes topological information in edge features for the classification of malicious flows. However, the graph $G$ of a large-scale IoT network is massive, which poses significant challenges to the timeliness of NIDS. Note that most existing GNN-based NIDSs [10, 17, 35, 37] are carried out in a centralized manner. These methods cause excessive computational complexity for GNN training and inference when applied to massive graphs, and therefore cannot be implemented IoT devices, including high-end ones. Therefore, the centralized realization of large-scale NIDS often rests on the Cloud to guarantee efficient processing of GNNs. Nevertheless, constructing massive graphs on Cloud tends to incur large communication overhead as NIDS must collect network flow data of the entire IoT network. Further considering the congested transmission of Internet backhaul to the remote Cloud, the delay for graph construction becomes intolerable.

This motivates us to propose MPGNN for establishing a parallel GNN-based NIDS. The key philosophy of MPGNN is to decompose the large-scale NIDS into multiple regional NIDSs whose communication overhead and computational complexity are manageable by high-end IoT devices. MPGNN follows a meta-learning framework and realizes parallel NIDS with two phases, namely *coalition formation* and *conditional intrusion detection*. During the coalition formation phase, MPGNN generates coalition formation policies to split the entire IoT network into multiple coalitions and selects a high-end IoT device for each generated coalition. The coalition head collects network flow features of devices within the coalition, and leverages GNN to establish a coalitional NIDS. These coalitional NIDSs operate in parallel, and work collaboratively to accomplish intrusion detection for large-scale IoT networks. In the following subsections, we give the details about coalition formation and coalitional intrusion detection.

### 3.2 Coalition Formation with Offline Graph Reinforcement Learning

The coalition formation policy generates coalitions based on the current state of IoT networks. Its output is a membership matrix $\mathbf{P} \in \mathbb{R}^{N \times K}$, where $N$ is the total number of devices in the IoT network and $K$ is the number of coalitions to be formed. The entry

$p_{n,k} \in \mathbb{P} \in [0, 1]$ denotes the inclination of including device $n$ in coalition $k$. The coalitions are formed based on the membership matrix $\mathbf{P}$ and a predetermined threshold $\delta$. If the inclination $p_{nk} \geq \delta$, then device $n$ is included in coalition $k$. Mathematically, coalitions can be defined by

$$C_k = \{n \mid p_{nk} \geq \delta, n \in \mathcal{N}, p_{nk} \in \mathbf{P}\}, k = 1, 2, \ldots, K. \quad (1)$$

Based on the above definitions, it is easy to see that a device can be associated with multiple coalitions. This overlap among coalitions is desirable in our problem for two reasons. Firstly, there is a possibility that certain devices contain critical patterns for identifying malicious network flows, and consequently, including these devices in multiple coalitions serves to improve the overall performance of coalitional intrusion detection. Secondly, IoT devices in the overlap between coalitions have the opportunity to leverage collaborative knowledge from multiple coalitional NIDSs through ensemble prediction. This presents a promising avenue for enhancing the stability of intrusion detection, particularly in the face of stealthy malicious attacks.

Before presenting the design of the coalition formation policy, it is essential to elucidate the impact of coalition formation on the performance of coalitional NIDS.

**How does coalition formation affect coalitional NIDS?** The coalition formation influences the timeliness and intrusion detection accuracy of coalitional NIDS. **1) Impact on Timeliness.** The timeliness of coalitional NIDS depends on the time complexity of GNN and the state of IoT networks. The time complexity of GNN-based coalitional NIDS is determined by the size of the formed coalitions. For example, the complexity of E-GraphSAGE for coalitional intrusion detection is $O(|C||\mathcal{E}_C|)$ [17] where $|C|$ is the number of devices in the coalition and $|\mathcal{E}_C|$ denotes the number of flows in the coalition. Given the fixed time complexity of GNN, the delay for running GNN-based NIDS is further affected by the states of IoT networks. Specifically, the delay for network flow data collection (which is required to construct the graph input to GNN-based NIDS) is determined by transmission rates between the coalition head and other members; the delay for GNN processing depends on the available computing resources of the coalition head. In particular, the network traffic incurred during NIDS interacts with the inherent dynamics of the IoT network, which forms the evolution of IoT network states and poses the need for periodic adjustments of collation. **2) Impact on Intrusion Detection Accuracy.** The effectiveness of GNN-based NIDS heavily relies on the quality and representativeness of network flow data utilized for training. However, the network flow data often exhibits substantial skewness. Consequently, the indiscriminate down-sampling of the entire IoT network may result in significant information losses about the malicious flows. To empirically substantiate this claim, we employ random graph sampling to generate coalitions on the NF-ToN-IoT-V2 [21] dataset. Fig. 2 shows the distribution of attack types in generated coalitions, which demonstrates that numerous types of malicious attacks are under-represented. This deficiency in representation adversely impacts the performance of coalitional NIDSs, as evidenced by the inferior intrusion detection performance depicted in Fig. 3.

It is difficult to precisely characterize the above impacts, especially considering the temporal dependency of coalition formations

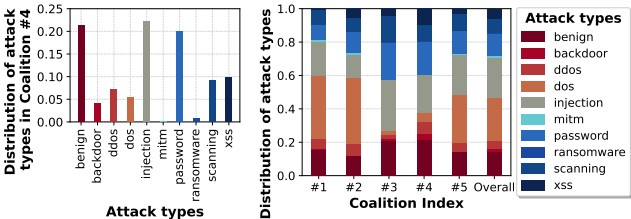

Figure 2: Distributions of attack types in random coalitions.

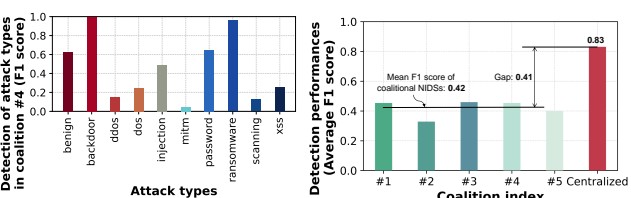

Figure 3: F1 score of coalitional NIDSs in random coalitions.

and IoT network states. Therefore, we propose a learning-based coalition formation policy. Particularly, we leverage the offline reinforcement learning framework to eliminate the overhead of collecting diverse interaction data. Next, we convert the coalition formation problem into a Markov decision process and propose an offline graph reinforcement learning algorithm as a solution.

**Coalition formation as a Markov decision process.** Due to the dynamic state of IoT networks, the coalition formation problem is formulated as a sequential decision-making process. The timeline is discretized into decision cycles, and a coalition formation decision is generated in each decision cycle. We note that the time scale of coalition formation (e.g., in hours) is much larger than that of coalitional intrusion detection (e.g., in milliseconds). The Markov Decision Process (MDP) for coalition formation can be defined by $(\mathcal{S}, \mathcal{A}, \mathcal{T}, R, \gamma)$, where $\mathcal{S}$ is the state space, $\mathcal{A}$ is the action space, $\mathcal{T}$ is the transition probability, $R$ is the reward function, and $\gamma \in [0, 1)$ is the time discount factor. At the beginning of decision cycle $t$, the learner observes state $\mathbf{s}_t \in \mathcal{S}$ of the IoT network, which may include the topological information, device energy state, device memory state, device bandwidth resource, etc. Then, the coalition formation policy $\pi : \mathcal{S} \rightarrow \mathcal{A}$ generates a decision $\mathbf{a}_t \in \mathcal{A}$ based on the observed state, expressed as $\mathbf{a}_t \leftarrow \pi(\mathbf{s}_t)$. Here, the $\mathbf{a}_t$ takes the form of membership matrix $\mathbf{P}$. The reward of decision $\mathbf{a}_t$ under $\mathbf{s}_t$ is determined by reward function $R$, i.e., $r_t = R(\mathbf{s}_t, \mathbf{a}_t)$. At the end of decision cycle $t$, the IoT network transits to a new state $\mathbf{s}_{t+1}$ based on the transition probability $\mathcal{T}$, i.e., $\mathbf{s}_{t+1} \sim \mathcal{T}(\cdot|\mathbf{s}_t, \mathbf{a}_t)$. In each decision cycle $t$, the objective is to maximize a sum of discounted rewards starting from $t$, i.e., $R_t = \sum_{\tau=t}^{\infty} \gamma^{\tau-t} r_\tau$. More details about the MDP for coalition formation are given in Appendix A.

Reinforcement learning (RL) serves as a potent tool for solving MDPs. However, RL algorithms typically operate in an online learning manner, which involves experience collection via iterative interactions with the environment. This process can be time-consuming, especially considering the large timescale of coalition formation.

Additionally, online RL algorithms spend considerable time in policy exploration, which can result in arbitrary poor performance for intrusion detection and leave IoT networks vulnerable to malicious attacks. To address these issues, MPGNN leverages offline graph reinforcement learning to establish a coalition formation policy based on historical data.

**Offline graph reinforcement learning** We propose an offline graph reinforcement learning method, namely Graph-Embedded Adversarially Trained Actor-Critic (G-ATAC), to learn a coalition formation policy. It utilizes ATAC [6] as a basic offline reinforcement learning framework, which guarantees *robust policy improvement*, i.e., maintaining safe policy improvement across large and anchored hyperparameter choices. G-ATAC incorporates GNN as the learning engine to effectively capture complicated topological information and dependencies of IoT devices.

G-ATAC operates on an offline dataset $\mathcal{D}$ that is collected by implementing arbitrary coalition formation policy (e.g., random formation) to IoT networks. A data sample $\mathbf{d} \in \mathcal{D}$ in historical dataset is denoted by $\mathbf{d} = (\mathbf{s}, \mathbf{a}, r, \mathbf{s}')$ where $r = R(\mathbf{s}, \mathbf{a})$, $\mathbf{s}' \sim \mathcal{T}(\cdot | \mathbf{s}, \mathbf{a})$. G-ATAC follows the Actor-Critic framework to handle the continuous action space of the membership matrix, and utilizes Graph Convolutional Network (GCN) to construct the Actor $\pi(\mathbf{s}; \Theta^\pi)$ and Critic $Q(\mathbf{s}, \mathbf{a}; \Theta^Q)$ for G-ATAC, where $\Theta^\pi$ and $\Theta^Q$ denote the trainable parameters of Actor and Critic, respectively. The Actor takes state $\mathbf{s}$ as input and directly outputs a membership matrix $\mathbf{a} \leftarrow \mathbf{P}$ for coalition formation. Using GCNs, the input state of actor $\mathbf{s}, \mathbf{s}' \in \mathcal{S}$ are written in the form of $\{\mathbf{A}, \mathbf{X}\}$, where $\mathbf{A}$ denotes the adjacent matrix with $a_{ij} \in \mathbf{A}$ characterizing the communication frequency between device $i$ and $j$; $\mathbf{X}$ represents device features (e.g., energy state, computing capacity, distribution of attacks, etc.) that may affect the coalition formation and the performance coalitional NIDSs. Suppose the GCN of Actor has $L$ Graph Convolutional Layers (GCLs), the propagation of $l$-th GCL can be expressed as:

$$\mathbf{H}^{(l+1)} = \sigma \left( \tilde{\mathbf{D}}^{-\frac{1}{2}} \tilde{\mathbf{A}} \tilde{\mathbf{D}}^{-\frac{1}{2}} \mathbf{H}^{(l)} \mathbf{W}^{(l)} \right) \tag{2}$$

where $\mathbf{H}^{(l)}$ is the matrix of activations in the $l$-th layer ($\mathbf{H}^{(0)}$ denotes the input node features $\mathbf{X}$); $\mathbf{W}^{(l)}$ is trainable weights in $l$-th layer; $\tilde{\mathbf{A}} = \mathbf{A} + \mathbf{I}_N$ is the weighted adjacent matrix with added self-loops with $\tilde{\mathbf{D}}_{ii} = \sum_j \tilde{\mathbf{A}}_{ij}$; $\sigma(\cdot)$ is the activation function. The output of the last GCL is the membership matrix, i.e., $\mathbf{P} \leftarrow \mathbf{H}^L$. Regarding Critic $Q(\mathbf{s}, \mathbf{a}; \Theta^Q)$, it employs the same architecture as the Actor. It takes $\mathbf{a} = \mathbf{P}$ and $\mathbf{s} = \{\mathbf{A}, \mathbf{X}\}$ as the input, and outputs the predicted Q-values that are used for updating the Actor.

We illustrate the learning process of G-ATAC in Fig. 4. G-ATAC completes training by following a Stackelberg Game [31], with Actor acting as Leader and Critic acting as Follower. This process can be expressed by the objectives below:

$$\pi^* \in \arg\max_{\pi \in \Pi} \mathcal{L}_{\mathcal{D}}(\pi, Q^\pi)$$
$$\text{s.t.} \quad Q^\pi \in \arg\min_{Q \in Q} \mathcal{L}_{\mathcal{D}}(\pi, Q) + \beta \mathcal{E}_{\mathcal{D}}(\pi, Q) \tag{3}$$

where $\mathcal{L}_{\mathcal{D}}(\pi, Q) = \mathbb{E}_{\mathcal{D}}[Q(\mathbf{s}, \pi) - Q(\mathbf{s}, \mathbf{a})]$ is the loss term related to relative pessimism, $\mathcal{E}_{\mathcal{D}}(\pi, Q) = \mathbb{E}_{\mathcal{D}}[(Q(\mathbf{s}, \mathbf{a}) - r - \gamma Q(\mathbf{s}', \pi))^2] - \min_{Q' \in Q} \mathbb{E}_{\mathcal{D}}[(Q'(\mathbf{s}, \mathbf{a}) - r - \gamma Q(\mathbf{s}', \pi))^2]$ is the term used to approximate Bellman consistency, and $\beta > 0$ is used to balance the importance of the above two terms. The objective of Actor is to

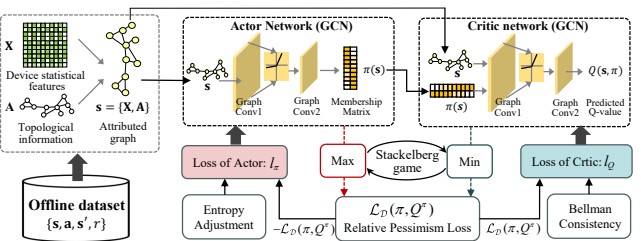

**Figure 4: The framework of G-ATAC.**

maximize the evaluation value assigned by the Critic, while Critic evaluates policy with a relatively pessimistic attitude.

To mitigate the deadly triad [27], $\mathcal{E}_{\mathcal{D}}$ combines temporal difference (td) losses of Critic and delayed targets [2], denoted as:

$$\mathcal{E}_{\mathcal{D}}^w = (1 - w)\mathcal{E}_{\mathcal{D}}^{\text{td}}(Q, Q, \pi) + w\mathcal{E}_{\mathcal{D}}^{\text{td}}(Q, \overline{Q}, \pi) \tag{4}$$

where $\mathcal{E}_{\mathcal{D}}^{\text{td}}(Q, Q', \pi) = \mathbb{E}_{\mathcal{D}}[(f(\mathbf{s}, \mathbf{a}) - r - \gamma Q'(\mathbf{s}'\pi))^2]$, $\overline{Q}$ is target network, and $w$ is the importance weight. Then, the loss of Critic can be expressed as $\ell_Q = \mathcal{L}_{\mathcal{D}}(\pi, Q) + \beta \mathcal{E}_{\mathcal{D}}^w(\pi, Q)$. To avoid the local optimality, Actor adopts the adaptive policy entropy adjustment [12]. Specifically, Actor adds a Lagrange relaxation term in its loss $\ell_\pi$. The Lagrange relaxation term is defined as $\alpha(\mathbb{E}_{\mathcal{D}}[\pi \log \pi] - \Delta_{\min})$, where $\mathbb{E}_{\mathcal{D}}[\pi \log \pi]$ is the entropy of the current policy, and $\Delta_{\min}$ is the expected minimum policy entropy, and $\alpha$ is an adaptive weight. In brief, the loss of Actor can be described as $\ell_\pi(\pi, \alpha) = -\mathcal{L}_{\mathcal{D}}(\pi, Q) - \alpha(\mathbb{E}_{\mathcal{D}}[\pi \log \pi] - \Delta_{\min})$. The training of Actor and Critic is realized by the Double Q Residual Algorithm (DQRA) [6]. The detailed training process of G-ATAC is provided in Algorithm 2, Appendix A.

## 3.3 Coalitional NIDS with E-GraphSAGE

Given formed coalitions in the previous phase, MPGNN proceeds to establish a coalitional NIDS within each coalition. Each coalition $C_k$ first chooses a high-end device (with joint optimization of computing capacity, energy resource, and communication efficiency) as the coalition head, denoted as $h_k$. The head is responsible for gathering and analyzing the real-time network flows, and performing coalitional intrusion detection within coalition $C_k$.

The flows in coalition $C_k$ are modeled as an attributed undirected multigraph, denoted as $G_{\text{NID},k} = (\mathcal{V}_{\text{ip},k} \times \mathcal{V}_{\text{port},k}, \mathcal{E}_{C_k}, \mathbf{E}_k)$. Specifically, the vertices of $G_{\text{NID},k}$ are identified by the flow endpoints, i.e., $\mathcal{V}_{\text{ip},k}$ (source IP, destination IP) and $\mathcal{V}_{\text{port},k}$ (source port, destination port), and the edge features $\mathbf{E}_k$ represent the statistical characteristics of network flows in $C_k$, e.g., the number of packets, number of bytes, flow duration, etc. It is essential to note that the vertex represents a port of an IoT device.

Given constructed attributed multigraphs, we convert the coalitional intrusion detection into an edge classification problem, then solve it with E-GraphSAGE [17]. E-GraphSAGE possesses a distinctive capability of analyzing edge features, making it well-suited for edge classification. The inputs to E-GraphSAGE are the set of edge features, denoted by $\{\mathbf{e}_{uv}, \forall uv \in \mathcal{E}_{C_k}\}$, and the set of node features, denoted by $\{\mathbf{x}_v, \forall v \in C_k\}$. E-GraphSAGE performs aggregation of edge features centered on the vertices in each layer. Consider a GNN with $L$ E-GraphSAGE layers, the aggregated embedding of

the neighborhood edges of the vertex $v$ in the $l$-th layer is:

$$\mathbf{h}^l_{\mathcal{N}(v)} = \mathrm{AGG}^l\left(\left\{\mathbf{e}^{l-1}_{uv}, \forall u \in \mathcal{N}(v), uv \in \mathcal{E}\right\}\right) \quad (5)$$

where $\mathbf{e}^{k-1}_{uv}$ is the feature embedding of edge $uv$ in the neighborhood $\mathcal{N}(v)$ in the $l-1$ layer, and $\mathrm{AGG}^l(\cdot)$ is the aggregation function of $l$-th layer. Our work utilizes mean function as the aggregation function, and therefore Eqn.(5) simplifies to $\mathbf{h}^l_{\mathcal{N}(v)} = \sum_{u \in \mathcal{N}(v), uv \in \mathcal{E}} \frac{\mathbf{e}^{l-1}_{uv}}{|\mathcal{N}(v)|_e}$, where $|\mathcal{N}(v)|_e$ denotes the number of edges in the neighborhood. After the aggregation of neighborhood edges, the embedding of node $v$ is updated as follows by connecting the previous layer node embedding $\mathbf{h}^{l-1}_v$ with the current neighborhood embedding $\mathbf{h}^l_{\mathcal{N}(v)}$:

$$\mathbf{h}^l_v = \sigma\left(\mathbf{W}^l \mathrm{concat}\left(\mathbf{h}^{l-1}_v, \mathbf{h}^l_{\mathcal{N}(v)}\right)\right) \quad (6)$$

where $\mathbf{W}^l$ is the trainable weight matrix of the $l$-th layer, concat represents concatenation, and $\sigma(\cdot)$ is the nonlinear activation function. The final node embedding of node $v$ is the output of the last layer, denoted as $\mathbf{z}_v = \mathbf{h}^L_v$. The final embedding of the edge $uv$ is obtained by concatenating the final node embeddings of the two endpoints, which can be denoted as $\mathbf{z}_{uv} = \mathrm{concat}(\mathbf{z}_u, \mathbf{z}_v)$.

To accomplish intrusion detection, it is necessary for the final edge embedding to go through a liner layer, which enables the acquisition of the predicted label for each edge. The loss function used for training the E-GraphSAGE model is defined as $\ell_{\mathrm{NID}} = H(\hat{Y}, Y)$, where $\hat{Y}$ is the predicted labels by E-GraphSAGE and $Y$ represents the true labels, and $H(\cdot)$ is the cross-entropy loss function. During the training process, the trainable parameters are optimized using an Adam optimizer. A detailed procedure for intrusion detection is provided in Algorithm 3 in Appendix A.

**Ensemble prediction with coalitional NIDSs.** Recall that the formed coalitions contain the overlap of IoT devices, we leverage ensemble prediction to combine the intrusion detection results of multiple coalitional NIDSs. Let $\hat{\mathbf{y}}_k(uv) = \Phi_k(uv)$ denotes the classification (i.e., intrusion detection) of edge $uv$ in coalition $k$, where $\Phi_k(\cdot)$ denotes the coalitional NIDS for coalition $k$ and $\hat{\mathbf{y}}_k(uv) = \{\hat{y}_{k,z}(uv)\}^Z_{z=1}$ denotes the confidence vector for $Z$ attack types. We let $\mathcal{K}(uv)$ denote the set of coalitions that contain $u$ and $v$, then the ensemble prediction of edge $uv$ is performed by coalitional NIDSs $\{\Phi_k\}_{k \in \mathcal{K}(uv)}$. The ensemble prediction result of edge $uv$ is:

$$\hat{\mathbf{y}}(uv) = \frac{1}{|\mathcal{K}(uv)|} \sum_{k \in \mathcal{K}(uv)} \hat{\mathbf{y}}_k(uv), \quad (7)$$

and the class of edge $uv$ is determined by $\arg\max_z \hat{y}_z(uv)$.

# 4 EXPERIMENT

## 4.1 Experiment Setting

**Datasets and Preprocessings.** We evaluate the performance of MPGNN on two real-world datasets: NF-ToN-IoT-V2 [21] and NF-CSE-CIC-IDS2018-V2[21]. The NF-ToN-IoT-V2 dataset comprises 10 categories (16,940,496 samples), including 9 types of attacks (63.99% of the total) and benign instances. The NF-CSE-CIC-IDS-2018-V2 dataset consists of 15 categories (18,893,708 samples), mapped into 7 classes, including 6 types of attacks (11.95% of the total) and benign samples. The detailed information on both datasets is available in Appendix B.1. The data preprocessing encompasses 1) imbalance

mitigation, 2) IoT device sampling, 3) training and test data preparation, and 4) offline datasets construction (for G-ATAC). Details about data preprocessing can be found in Appendix B. After the preprocessing, the two datasets contain 252,020 and 348,485 endpoints, respectively.

**Baselines and Metrics.** We compare MPGNN with centralized NIDS and other parallel realizations with commonly used network partitioning strategies, including random partitioning, Louvain partitioning [4], FastGreedy partitioning [8], and DL-based partitioning [18]. MPGNN, random partitioning, and DL-based partitioning divide the network into five coalitions, whereas Louvain partitioning and FastGreedy partitioning automatically determine the optimal number of coalitions.

The performance of MPGNN is assessed from two perspectives: intrusion detection performance and complexity. Specifically, four metrics, i.e., accuracy (ACC), precision (PRE), recall (REC), F1 score (F1), are used to evaluate the performance of intrusion detection. Two metrics, i.e., Training Time (TT) and Memory Size (SIZE), are used to the complexity of MPGNN. Besides, to assess the scale of coalitions, we introduce the average number of nodes (AN) of coalitions as another evaluation metric. Detailed definitions of the aforementioned metrics are provided in Appendix B.

## 4.2 Results and Discussions

**Comparison with baselines.** The performance of MPGNN and baselines are summarized in Table 1. Intuitively, the centralized NIDS gives the highest intrusion detection accuracy as it avoids information losses. Among all parallel realizations, MPGNN achieves the best performance for intrusion detection, providing an average accuracy of 83.39% and 90.28% on two datasets, respectively, aligning closely with the performance of the centralized NIDS (i.e., 84.59% and 92.42%). We see that MPGNN-MLP has a noticeable performance decline on both datasets compared to MPGNN, which justifies the efficacy of using GNNs in coalition formation. However, the detection accuracy of MPGNN-MLP still far outperforms the other traditional graph partitioning schemes. This indicates that utilizing learning-based strategies for coalition formation is beneficial.

Moreover, in comparison with the centralized model, MPGNN achieves a remarkable 41.63% (22.11%) reduction in training time and a 15.06% (8.53%) decrease in memory consumption on the NF-ToN-IoT-V2 (NF-CSE-CIC-IDS2018-V2) dataset. These findings indicate the effectiveness of MPGNN for computational burden optimization and the potential for deployment at high-end IoT devices. Additionally, the average coalition size of MPGNN is 61 nodes and 62.2 nodes, respectively, a significantly reduced scale when compared with that of the centralized network. This disparity in scale underscores the efficacy of MPGNN in mitigating communication overhead and computational complexity.

Fig.5 depicts the data distribution of attack types for coalitions generated by Random, FastGreedy, and MPGNN on the NF-ToN-IoT-V2 dataset. (The corresponding results on the NF-CSE-CIC-IDS-2018-V2 dataset are available in Fig.12). It can be observed in Fig. 5 that the data distribution of coalitions generated by Random and FastGreedy exhibits a pronounced imbalance, causing the exclusion of certain types of attacks. This imbalance may lead to inferior

**Table 1: Performance of MPGNN and baselines.**

| Dataset | Method | ACC | REC | PRE | F1 | TT(s) | SIZE(MB) | AN |
|---|---|---|---|---|---|---|---|---|
| NF-ToN-IoT-V2 | Global | 84.59% | 0.88 | 0.81 | 0.83 | 772.48 | 181.00 | 200.0 |
| | MPGNN | 83.39% | **0.86** | **0.80** | **0.82** | 450.86 | 153.75 | 61.0 |
| | MPGNN-MLP[18] | 78.11% | 0.81 | 0.79 | 0.77 | 389.49 | 146.60 | 125.2 |
| | Random | 51.06% | 0.48 | 0.50 | 0.42 | 309.24 | 108.55 | 71.2 |
| | Louvain [4] | 28.99% | 0.23 | 0.22 | 0.17 | **120.45** | **39.07** | **28.6** |
| | FastGreedy [8] | 55.03% | 0.50 | 0.43 | 0.44 | 274.75 | 95.88 | 66.7 |
| NF-CSE-CIC-IDS2018-V2 | Global | 92.42% | 0.86 | 0.80 | 0.79 | 583.87 | 250.85 | 200.0 |
| | MPGNN | 90.28% | **0.83** | **0.80** | **0.78** | 454.76 | 229.46 | 62.2 |
| | MPGNN-MLP [18] | 85.10% | 0.80 | 0.74 | 0.72 | 417.02 | 188.99 | 95.8 |
| | Random | 61.13% | 0.59 | 0.22 | 0.49 | 274.98 | 134.95 | 71.2 |
| | Louvain [4] | 23.11% | 0.22 | 0.14 | 0.12 | 150.86 | 57.24 | 40.0 |
| | FastGreedy [8] | 18.31% | 0.23 | 0.08 | 0.11 | **85.39** | **38.97** | 33.3 |

detection performance. In contrast, the coalitions generated by MPGNN exhibit a balanced distribution of attack types, which contributes to the performance enhancement in intrusion detection.

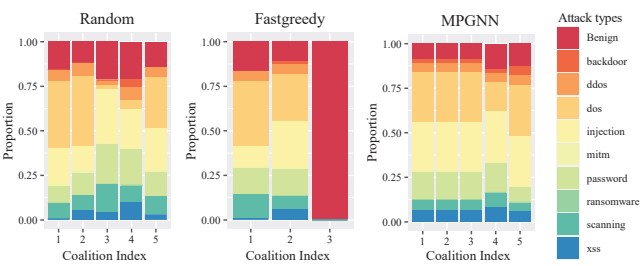

**Figure 5: Distribution of attack types within coalitions**

**Coalition size.** Fig. 6 shows the size of coalitions generated by MPGNN and other parallel realizations on NF-ToN-IoT-V2. As aforementioned, the coalition size directly affects the communication and computation overheads of coalition NIDS. To guarantee the efficiency of coalitional NIDS, the coalition size should not be kept low. We can see from Fig. 6 that the average coalition size of MPGNN is 61.0, providing a scale reduction of 3.3× compared to the original IoT network (with the size of 200). However, it is important to note that solely focusing on reducing the communication and computation overhead may lead to performance degradation in intrusion detection. For instance, the Louvain method produces an average coalition size of 28.57, which is too small to provide sufficient data for training a well-performing coalitional NIDS. Additionally, by analyzing the results presented in Table 1 and Fig. 6, we can infer that allowing appropriate overlap among coalitions helps improve the performance of intrusion detection. However, the degree of overlapping should be judiciously determined, otherwise, it may lead to a large coalition size that harms the efficiency of coalitional NIDS, e.g., MPGNN-MLP.

**Performance breakdown to coalitions and attack types.** We then provide a comprehensive assessment of MPGNN by breaking down its intrusion detection performance to each formed coalition and each type of attack. Given the performance breakdown for NF-ToN-IoT-V2 in Table 2, we see that the intrusion detection performances of the five coalitional NDIS are similar to each other, with macro average around 0.86 and F1-scores around 0.81. In terms of specific attack types, the average F1-scores for Backdoor, DoS, Ransomware, and Scanning with all five coalitional NIDSs are high,

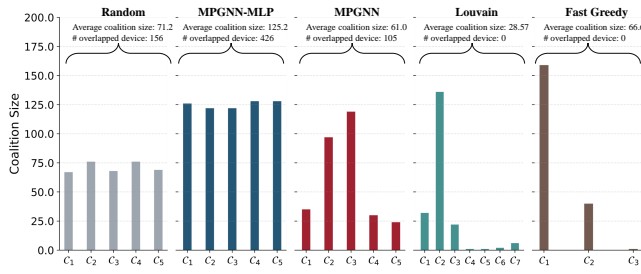

**Figure 6: Comparison of coalition size.**

**Table 2: Performance breakdown (NF-ToN-IoT-V2)**

| Metrics | Coalition1 | | Coalition2 | | Coalition3 | | Coalition4 | | Coalition5 | | Average | |
|---|---|---|---|---|---|---|---|---|---|---|---|---|
| | REC | F1 | REC | F1 | REC | F1 | REC | F1 | REC | F1 | REC | F1 |
| Benign | 0.84 | 0.88 | 0.85 | 0.91 | 0.85 | 0.91 | 0.87 | 0.92 | 0.87 | 0.91 | 0.86 | 0.91 |
| Backdoor | 1.00 | 1.00 | 1.00 | 1.00 | 1.00 | 1.00 | 1.00 | 0.99 | 1.00 | 0.99 | 1.00 | 1.00 |
| DoS | 0.93 | 0.92 | 0.94 | 0.93 | 0.90 | 0.92 | 0.91 | 0.93 | 0.95 | 0.93 | 0.93 | 0.93 |
| DDoS | 0.92 | 0.75 | 0.90 | 0.80 | 0.95 | 0.67 | 0.94 | 0.77 | 0.93 | 0.74 | 0.93 | 0.75 |
| Injection | 0.69 | 0.78 | 0.64 | 0.75 | 0.65 | 0.76 | 0.65 | 0.76 | 0.58 | 0.71 | 0.64 | 0.75 |
| MITM | 0.59 | 0.53 | 0.65 | 0.50 | 0.61 | 0.53 | 0.58 | 0.54 | 0.65 | 0.49 | 0.62 | 0.52 |
| Password | 0.87 | 0.81 | 0.88 | 0.77 | 0.84 | 0.81 | 0.89 | 0.81 | 0.87 | 0.76 | 0.87 | 0.79 |
| Ransomware | 0.98 | 0.97 | 0.99 | 0.98 | 0.99 | 0.97 | 1.00 | 0.98 | 0.97 | 0.96 | 0.99 | 0.97 |
| Scanning | 0.97 | 0.92 | 0.97 | 0.94 | 0.97 | 0.95 | 0.97 | 0.95 | 0.96 | 0.95 | 0.97 | 0.94 |
| XSS | 0.77 | 0.65 | 0.77 | 0.64 | 0.85 | 0.62 | 0.88 | 0.64 | 0.75 | 0.63 | 0.80 | 0.64 |
| **Macro Avg.** | 0.85 | 0.82 | 0.86 | 0.82 | 0.86 | 0.81 | 0.87 | 0.83 | 0.85 | 0.81 | 0.86 | 0.82 |

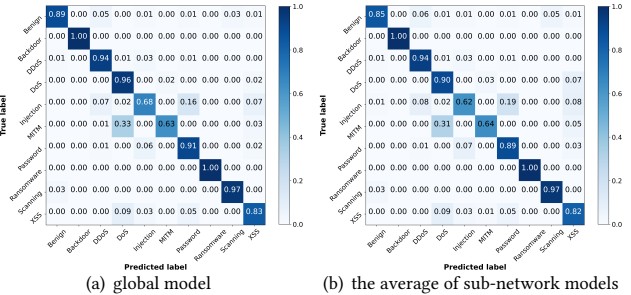

(a) global model     (b) the average of sub-network models

**Figure 7: Normalized confusion matrix on NF-ToN-IoT-v2**

around 1.00, 0.93, 0.97, and 0.94, respectively. These attack types exhibit prominent spatial correlations, making them easily detectable with GNN. Fig. 7 further provides normalized confusion matrices for centralized NIDS and MPGNN. It can be observed that the detection performance of each attack type achieved by MPGNN closely approximates that of centralized NIDS.

Table 3 shows the performance breakdown of MPGNN on the NF-CSE-CIC-IDS2018-V2 dataset. The average recall and average F1-score of five coalitional NIDSs generated by MPGNN are 0.83 and 0.78, respectively. Notably, the recall and F1-score for DoS, Bot, and BruteForce stand out high, approaching 1.00. This is due to the distinct spatial characteristics and adequate training samples of these attack types. In contrast, the recall and F1 score for Web attack are low, because the data samples of Web attack are limited (3502 samples). This implies that the insufficient sample size may become a performance bottleneck of MPGNN.

**Device capacities within coalitions.** Fig. 8 depicts the computing resource of IoT devices within coalitions generated by MPGNN.

**Table 3: Performance breakdown (NF-CSE-CIC-IDS2018-V2)**

| Metrics | Coalition1 | | Coalition2 | | Coalition3 | | Coalition4 | | Coalition5 | | Average | |
|---|---|---|---|---|---|---|---|---|---|---|---|---|
| | REC | F1 | REC | F1 | REC | F1 | REC | F1 | REC | F1 | REC | F1 |
| Benign | 0.96 | 0.64 | 0.94 | 0.71 | 0.93 | 0.70 | 0.96 | 0.72 | 0.82 | 0.56 | 0.92 | 0.67 |
| BruteForce | 1.00 | 1.00 | 1.00 | 1.00 | 1.00 | 1.00 | 1.00 | 1.00 | 1.00 | 1.00 | 1.00 | 1.00 |
| Bot | 1.00 | 1.00 | 1.00 | 1.00 | 1.00 | 1.00 | 1.00 | 1.00 | 1.00 | 1.00 | 1.00 | 1.00 |
| DoS | 1.00 | 1.00 | 1.00 | 1.00 | 1.00 | 1.00 | 1.00 | 1.00 | 1.00 | 1.00 | 1.00 | 1.00 |
| DDoS | 0.93 | 0.96 | 1.00 | 0.99 | 0.71 | 0.83 | 0.84 | 0.91 | 0.98 | 0.99 | 0.89 | 0.94 |
| Infiltration | 0.36 | 0.52 | 0.56 | 0.70 | 0.55 | 0.69 | 0.55 | 0.70 | 0.31 | 0.44 | 0.47 | 0.61 |
| Web Attacks | 0.47 | 0.13 | 0.46 | 0.54 | 0.77 | 0.06 | 0.62 | 0.08 | 0.48 | 0.34 | 0.56 | 0.23 |
| **Macro Avg.** | 0.82 | 0.75 | 0.85 | 0.85 | 0.85 | 0.75 | 0.85 | 0.77 | 0.80 | 0.76 | 0.83 | 0.78 |



**Figure 8: Visualization of device capacities in coalitions.**

**Table 4: Results of ablation experiments**

| Dataset | Experimental Setting | ACC | REC | PRE | F1 | TT(s) | SIZE(MB) | AN |
|---|---|---|---|---|---|---|---|---|
| NF-ToN-IoT-V2 | MPGNN | 83.39% | 0.86 | 0.80 | 0.82 | 450.86 | 153.75 | 61.0 |
| | MPGNN-MLP | 78.11% | 0.81 | 0.79 | 0.77 | 389.49 | 146.6 | 125.2 |
| | MPGNN-unweighted A | 44.69% | 0.54 | 0.46 | 0.43 | 212.11 | 76.67 | 44.2 |
| | MPGNN w.o. DQRA | 82.81% | 0.83 | 0.76 | 0.78 | 481.31 | 176.32 | 57.6 |
| NF-CSE-CIC-IDS2018-V2 | MPGNN | 90.28% | 0.83 | 0.80 | 0.78 | 454.76 | 229.46 | 62.2 |
| | MPGNN-MLP | 85.10% | 0.80 | 0.74 | 0.72 | 417.0237 | 188.99 | 95.8 |
| | MPGNN-unweighted A | 40.93% | 0.44 | 0.31 | 0.34 | 226.54 | 106.85 | 57.0 |
| | MPGNN w.o. DQRA | 88.70% | 0.80 | 0.82 | 0.76 | 451.12 | 225.97 | 61.8 |

The nodes denote the IoT devices, and the size of the nodes is proportional to the amount of computing resources. The links denote the communication connections between IoT devices. Nodes in different coalitions are assigned with different colors. Based on the result in Fig. 8, we see that high-end devices are assigned to coalitions in a balanced manner. It can be observed that each coalition includes at least one high-end device that is capable of running coalitional NIDS.

**Ablation experiments.** We further conduct ablation experiments to evaluate the efficacy of designing elements of MPGNN. The experimental results are summarized in Tabel 4. MPGNN-MLP replaces the GNN in coalition formation policy (i.e., offline graph reinforcement learning) with MLP. We see from Table 4 that MPGNN-MLP leads to a noticeable decline in intrusion detection performance on both datasets. Additionally, MPGNN increases the average coalition size, which increases the computational complexity of coalitional NIDS. MPGNN-unweighted **A** substitutes the weighted adjacent matrix with a binary matrix to conduct a confirmatory experiment. As shown in Table 4, for both datasets, neglecting the interaction patterns between nodes causes severe performance degradation in intrusion detection, as this process loses the topological information of IoT networks. MPGNN w.o. DQRA removes the DQRA mechanism in the training process of G-ATAC. The experimental results show a slight decrement in the intrusion detection performance. This is because the absence of DQRA may cause overestimation during training.

$\beta$ is the hyperparameter used to balance the importance of relative pessimism and Bellman consistency in Eqn. (3). Table 5 presents the impact of $\beta$ values on the performance of intrusion detection.

**Table 5: Intrusion detection performance with different $\beta$**

| Dataset | $\beta$ value | ACC | REC | PRE | F1 |
|---|---|---|---|---|---|
| NF-ToN-IoT-V2 | $\beta = 1$ | 58.41% | 0.5949 | 0.5419 | 0.5353 |
| | $\beta = 4$ | **83.39%** | **0.8581** | **0.8014** | **0.8187** |
| | $\beta = 16$ | 77.19% | 0.7730 | 0.7303 | 0.7270 |
| | $\beta = 64$ | 73.30% | 0.7574 | 0.7200 | 0.7052 |
| NF-CSE-CIC-IDS2018-V2 | $\beta = 1$ | 78.87% | 0.6995 | 0.6351 | 0.6141 |
| | $\beta = 4$ | 90.28% | 0.8337 | 0.8039 | 0.7769 |
| | $\beta = 16$ | 81.89% | 0.7529 | 0.7139 | 0.6943 |
| | $\beta = 64$ | **90.34%** | **0.8462** | **0.8109** | 0.7836 |

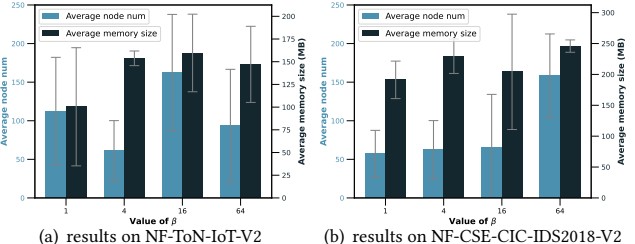

(a) results on NF-ToN-IoT-V2    (b) results on NF-CSE-CIC-IDS2018-V2

**Figure 9: The mean and standard deviation of node numbers and memory size with different $\beta$**

The coalitional NIDSs achieve optimal performance with $\beta = 16$ on the NF-ToN-IoT-V2 dataset, and with $\beta = 64$ on the NF-CSE-CIC-IDS2018-V2 dataset. Fig. 9 visualizes the mean and standard deviation of node numbers and memory size across coalitions with varying $\beta$ values. We see that the choice of $\beta$ value significantly impacts the scale and the memory consumption of coalitions. In Fig. 9, on the NF-CSE-CIC-IDS2018-V2 dataset, $\beta = 64$ results in a considerably larger average coalition scale compared to other $\beta$ values, potentially explaining the improvement in detection performance.

## 5 CONCLUSION

This paper proposed MPGNN to provide a parallel framework for GNN-based NIDS in large-scale IoT networks. MPGNN leverages a meta-learning framework that learns a coalition formation policy as meta-knowledge to divide the large-scale IoT network into several loosely coupled subnetworks (i.e., coalitions), and then performs coalitional network intrusion detection with reduced communication overhead and computational complexity manageable high-end devices in the IoT network. We proposed an offline graph reinforcement learning method to obtain a coalition formation policy. Finally, MPGNN utilizes E-GraphSAGE to establish coalitional NIDSs, which work collaboratively via ensemble prediction to accomplish intrusion detection for large-scale IoT networks. We evaluated the proposed method on two real-world datasets and compared it with the centralized NIDS and parallel realizations supported by Random, Louvain, and FastGreedy. The experimental results validate the efficacy of MPGNN. While MPGNN is presented for network intrusion detection for large IoT networks, its potential extends to other scenarios that involve judicious parallelism of graph neural networks.

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

## A   APPENDIX: DETAILS OF META PARALLEL GRAPH NEURAL NETWORKS (MPGNN)

### A.1   State space, transition process, and reward function for reinforcement learning

**State Space**. Recall that the state space is denoted by $\mathcal{S}$. Because G-ATAC uses GNNs as its learning engine. A state $\mathbf{s} \in \mathcal{S}$ can be represented by $\mathbf{s} = \{\mathbf{A}, \mathbf{X}\}$. $\mathbf{A} \in \mathbb{R}^{M \times M}$ is a weighted adjacent matrix where $a_{ij}$ is the frequency that device $i$ transmits messages to device $j$. $\mathbf{X} = \{\mathbf{x}_i\}_{i \in \mathcal{N}}$ collects the statistical features of IoT devices in $\mathcal{N}$ where $\mathbf{x}_i$ denotes the features of IoT device $i$. In our problem, $\mathbf{x}_i$ may contain 1) The flow features associated with the device. These features include the in-degree and out-degree of network flows and the one-hot encoding of distributions of attack types, which can affect the performance of coalitional NIDS; 2) The energy state of IoT devices. An IoT device can be connected to the power grid or relies on the battery. The energy state affects the selection of coalition head, for example, a device with low battery levels cannot accomplish coalitional intrusion detection. 3) Computing capacity of devices. The computing capacity determines the delay for running GNN-based intrusion detection. 4) The memory state of IoT devices. The coalition head should have enough memory to load GNNs for running coalition NIDS. 4) The communication bandwidth of IoT devices. The communication bandwidth affects the delay for the coalition head to collect flow features within the coalition.

**Transition.** 1) Transition of Energy state. For IoT devices connected to the power grid, their battery states are always set as high. For IoT devices that use batteries, their energy state is determined by the transition as $b' = b - \int_{t_0}^{t_0+t} I(t_0)dt$, where $b$ and $b'$ represent the current energy state and next energy state, respectively, and $I(t_0)$ denotes the instantaneous current consumed by the circuit of the IoT node at the specific time $t_0$. In the experiment, the coalition head incurs large instantaneous currents for running coalition NIDSs, and other IoT devices only incur low instantaneous currents for supporting routine activities. Regarding the communication state, we consider that the IoT devices use frequency division multiplexing, and therefore bandwidth consumption is proportional to the number of nodes it communicates. The state transition of the bandwidth can be represented as $c'_i = c_i - \epsilon_c \cdot n_i$, where $\epsilon_c$ is a coefficient of scaling, and $n_i$ is the number of nodes with which it communicates.

**Reward Function**. The reward function for G-ATAC is defined as $r = r_{\mathrm{NID}} - w_a \cdot p_a - w_c \cdot p_c - w_b \cdot p_b$, where $r_{\mathrm{NID}} = \Sigma_{k=1}^{K}(\mathrm{F1}_k + \mathrm{ACC}_k)$ evaluates the intrusion detection effectiveness for each coalition, with $K$ denoting the number of coalitions generated by G-ATAC; $\mathrm{F1}_k$ and $\mathrm{ACC}_k$ are the F1-score and accuracy of the intrusion detection model in the $k$-th coalition, respectively.

The penalty term $p_a$ is related to the number of sample categories, defined by $p_a = \Sigma_{k=1}^{K} \frac{Z-A(k)}{Z}$. Here, $A(k)$ represents the count of attack types contained in the $k$-th coalition, while $T$ denotes the total number of attack types in the dataset. $p_c$ serves as the penalty term aimed at mitigating communication overhead by constraining the number of nodes within each coalition. Its formulation is given by $p_c = \Sigma_{k=1}^{K} \delta_c(k) \frac{C_k - \xi}{\xi}$, where $\xi$ signifies the upper threshold for the expected scale of coalitions, $C_k$ denotes the number of devices in coalition $C_k$, and $\delta(k)$ indicates whether the scale of the $k$-th coalition surpasses the predefined threshold. Lastly, $p_b$ serves as the penalty term employed to evaluate the adequacy of energy and memory resources within selected coalition heads. It is defined as $p_b = \Sigma_{k=1}^{K} \delta_b(k)$, where $\delta_b(i)$ evaluates whether the energy or memory component in $\mathbf{s}'$ of $h_k$ (i.e., the coalition head in $C_k$) is negative. Additionally, $w_a$, $w_c$, and $w_b$ represent the weights that determine the respective contributions of $p_a$, $p_c$, and $p_b$ within the reward function. In the experiment, the $w_a$ is set to 1, while $w_c$ is set to 5, and $w_b$ is set to 50.

### A.2   Pseudocodes of MPGNN

The overall framework of MPGNN is available in Algorithm 1. The training of G-ATAC is given is shown in Algorithm 2. The training of E-GraphSAGE is given in Algorithm 3.

## B   APPENDIX: DETAILS ABOUT EXPERIMENTAL SETTINGS

### B.1   Detailed Information of Intrusion Detection Datasets

Two recently released public datasets namely NF-ToN-IoT-V2 and NF-CSE-CIC-IDS2018-V2 are used in this paper, whose detailed information is collated in Table 6. The NF-ToN-IoT-V2 dataset is derived from a well-known NIDS dataset namely ToN-IoT [20], which focuses specifically on IoT networks. Out of the entire number of data flows, which amounts to 16,940,496, 10,841,027 (63.00%) are abnormal samples while 6,099,469 (36.01%) are labeled as benign. The NF-ToN-IoT-V2 dataset consists of 10 distinct categories of samples, which include 9 forms of attacks as well as benign instances. In total, the dataset includes 43 features, which are associated with respective labels. The NF-CSE-CIC-IDS2018-V2 dataset is a NetFlow-based variant of CSE-CIC-IDS2018 [20] dataset, which contains network traffic collected over 10 days. There are 18,893,708 flows in total, of which 2,258,141 (11.95%) are attack samples and 16,635,567 (88.05%) are benign ones. Additionally, the NF-CSE-CIC-IDS2018-V2 dataset shares the same feature set with the NF-ToN-IoT-V2 dataset.

### B.2   Data Preprocessing

The process of data preparation primarily consists of four steps. Firstly, to alleviate the extreme imbalance of data, we proceed by sampling the two datasets based on their respective attack distributions. Then, the IP addresses within each dataset are arranged in descending order according to the frequency of their occurrence in both the source IPs and the destination IPs, while the top 200 IPs form a set, denoted as $\mathcal{P}$. Only the data flows whose source IP and destination IP both exist in $\mathcal{P}$ are retained. In practice, it's observed that this strategy does not

---

**Algorithm 1** The overall framework of MPGNN

---

**Input:**

 Device set $\mathcal{N} = \{1, 2, \ldots, N\}$; The number of coalitions to be formed $K$; Test graph constructed by test set $G\_test$; Coalition formulation policy set $\{\pi'_1, \pi'_2, \ldots\}$; Threshold for coalition formation $\delta$; A buffer to storage historical data instances $BUFFER = \varnothing$; Learning Rate $LR_\pi, LR_Q, LR_N$

**Output:** Detection results for the entire IoT network $\hat{Y}$

1: Network Context Extraction : $\mathbf{s} = \{\mathbf{A}, \mathbf{X}\}$
 // *Offline dataset construction*
2: Partition the large-scale IoT network by arbitrary policy:
  $\forall \pi' \in \{\pi_1, \pi_2, \ldots\}, \mathbf{P}' \leftarrow \pi'(s), \mathbf{a} \leftarrow \mathbf{P}',$
3: Get coalitions based on $\mathbf{a}'$:
  $C'_k = \{i \mid p_{ik} \geq \delta, i \in \mathcal{N}, p_{ik} \in \mathbf{P}'\}, k = 1, 2, \ldots, K$
4: Train coalitional NIDSs in $C'_k, k = 1, 2, \ldots, K$ in parallel :
  $Detection(G'_{\mathrm{NID}_k}; \hat{\theta}_k) \leftarrow \mathrm{ADAM}(\theta_k, LR_N)$
5: Calculate reward and state transition, then save instances in buffer:
  $r \leftarrow R(\mathbf{s}, \mathbf{a}), \mathbf{s}' \sim \mathcal{T}(\cdot|\mathbf{s}, \mathbf{a}) \quad BUFFER.append(\{\mathbf{s}, \mathbf{a}, r, \mathbf{s}'\})$
 // *Generation of coalition formation policy*
6: Offline graph reinforcement learning:
  $\pi(\mathbf{s}; \hat{\Theta}^\pi) \leftarrow \mathrm{ADAM}(\Theta^\pi, LR_\pi), Q(\mathbf{s}, \mathbf{a}; \hat{\Theta}^Q) \leftarrow \mathrm{ADAM}(\Theta^Q, LR_Q)$   **(More details are available in Appendix 2)**
7: Get the final coalition formation:
  $\mathbf{P} \leftarrow \pi(\mathbf{s}; \hat{\Theta}^\pi), C_k = \{i \mid p_{ik} \geq \delta, i \in \mathcal{N}, p_{ik} \in \mathbf{P}\}, k = 1, 2, \ldots, K$
 // *Coalitional intrusion detection*
8: Train the coalitional NIDSs in parallel:
 Detection$(G_{\mathrm{NID}_k}; \tilde{\theta}_k) \leftarrow ADAM(\theta_k, LR_N)$   **(More details are available in Appendix 3)**
9: Employ $Detection(\tilde{\theta}_i)$ for real-time detection:
  $\hat{Y}_i \leftarrow Detection(G_{test}, \tilde{\theta}_i)$
10: Realize real-time intrusion detection of global network via ensemble prediction: $\hat{Y} \leftarrow \{\hat{Y}_1, \hat{Y}_2, \ldots, \hat{Y}_K\}$
11: **return** Detection results for the entire IoT network $\hat{Y}$

---

**Table 6: Details of NF-ToN-IoT-V2 and NF-CSE-CIC-IDS2018-v2.**

| Dateset | NF-ToN-IoT-V2 | | NF-CSE-CIC-IDS2018-v2 | |
|---|---|---|---|---|
| **Total Number of Flows** | 16,940,496 | | 18,893,708 | |
| **Benign Flows Percentage** | 36.01% | | 88.05% | |
| | Class | Count | Class | Count |
| | Backdoor | 16,809 | BruteForce | 120,912 |
| | DoS | 712,609 | Bot | 143,097 |
| | DDoS | 2,026,234 | | |
| **Distribution** | Injection | 684,465 | DoS | 483,999 |
| | MITM | 7,723 | | |
| **of Attacks** | Password | 1,153,323 | DDoS | 1,390,270 |
| | Ransomware | 3,425 | Infiltration | 116,361 |
| | Scanning | 3,781,419 | Web Attacks | 3,502 |
| | XSS | 2,455,020 | | |

cause too much loss of samples. Thirdly, the sampled dataset is split into a train set and a test set with a ratio of 0.7:0.3, while ensuring that the samples in both sets are identically distributed.

Finally, it is important to construct the dataset for offline graph reinforcement learning. Various policies (e.g., random policy, expert policy, extreme policy, etc.) are employed to partition the global network into multiple overlapped coalitions according to the network states (e.g., energy condition, memory condition, etc.). Correspondingly, the data flows whose source IP or destination IP fell within each coalition are constructed as an attributed graph, which is used to train corresponding coalitional NIDS in parallel. The reward is calculated based on the detection performance and resource consumption. In the end, the tuples consisting of states, actions, rewards, and next states are stored.

The size of datasets for offline reinforcement learning datasets spans from several thousand to tens of millions. In our experiment, we find that a dataset comprising several thousand samples suffices for the training of coalition formation. We have constructed two datasets

**Algorithm 2** G-ATAC

**Input:**

    Offline dataset for G-ATAC $\mathcal{D}$; Loss term relative weight $\beta$; Target network update weight $\tau$; Lower bound of policy entropy $\Delta_{\min}$;
    Threshold $\delta$; Initialized adaptive weight $\alpha$; Training epoch $T$; Actor learning rate $LR_\pi$; Critic learning rate $LR_Q$; $\alpha$ learning rate $LR_\alpha$

**Output:**  Overlapped coalition set $\{C_1, C_2, \dots\}$

    *// Training of coalition formation policy*

  1: Initialized target network: $\overline{Q}_1 \leftarrow Q_1, \overline{Q}_2 \leftarrow Q_2$;

  2: **for** $t = 1, 2, ..., T$ **do**

  3:    Sample minibatch from $\mathcal{D}$: $\mathcal{D}_{\mathsf{mb}} \leftarrow \{\mathbf{s}, \mathbf{a}, r, \mathbf{s}'\}$

       *// Updata Critic:*

  4:    for $Q_i, i = 1, 2$:

$$\mathcal{L}_{\mathcal{D}_{\mathsf{mb}}} = \mathbb{E}_{\mathcal{D}_{\mathsf{mb}}}\left[Q_i(\mathbf{s}, \pi) - Q_i(\mathbf{s}, \mathbf{a})\right]$$
$$\mathcal{E}_{\mathcal{D}_{\mathsf{mb}}} = \tfrac{1}{2}\mathcal{E}^{td}_{\mathcal{D}_{\mathsf{mb}}}(Q_i, Q_i, \pi) + \tfrac{1}{2}\mathcal{E}^{td}_{\mathcal{D}_{\mathsf{mb}}}(Q_i, \overline{Q}_{\min}, \pi)$$
$$\ell_Q(Q_i) = \mathcal{L}_{\mathcal{D}_{\mathsf{mb}}}(\pi, Q_i) + \beta\mathcal{E}_{\mathcal{D}_{\mathsf{mb}}}(\pi, Q_i)$$
$$Q_i(\hat{\Theta}_i^Q) \leftarrow \text{ADAM}(\Theta_i^Q, LR_Q)$$

       *// Updata Actor:*

  5:    $\ell_\pi(\pi, \alpha) = -\mathcal{L}_{\mathcal{D}_{\mathsf{mb}}}(\pi, Q_1) - \alpha\left(\mathbb{E}_{\mathcal{D}_{\mathsf{mb}}}\left[\pi \log \pi\right] - \Delta_{\min}\right)$

$$\pi(\hat{\Theta}^\pi) \leftarrow \text{ADAM}(\Theta^\pi, LR_\pi)$$
$$\alpha \leftarrow \text{ADAM}(\alpha, LR_\alpha), \quad \alpha = \max\{0, \alpha\}$$

       *// Updata target network*

  6:    for $(Q_i, \overline{Q}_i)_{i=1,2}$:

$$\overline{Q}_i \leftarrow (1 - \tau)\overline{Q}_i + \tau Q_i$$

  7: **end for**

    *// Formation of coalitions*

  8: Get membership matrix $\mathbf{P} \leftarrow \pi(\mathbf{s}; \hat{\Theta}^\pi)$

  9: **for** $k = 1, 2, ...$ **do**

 10:    $C_k = \{n \mid p_{nk} \geq \delta, n \in \mathcal{N}, p_{nk} \in \mathbf{P}\}, k = 1, 2, \dots, K$

 11: **end for**

 12: **return**  Coalition set $\{C_1, C_2, \dots, C_K\}$;

for offline reinforcement learning-based coalition formation based on NF-ToN-IoT-V2 and NF-CSE-CIC-IDS2018-v2, which are denoted as ToN-Partition and CSE-Partition, respectively.

The ToN-Partition dataset contains a total of 2296 samples, whose distributions of reward and penalty terms are present in Fig. 10. Concurrently, the CSE-Partition dataset consists of 1840 samples. In this dataset, the distributions of rewards and related penalty terms are displayed in Fig. 11.

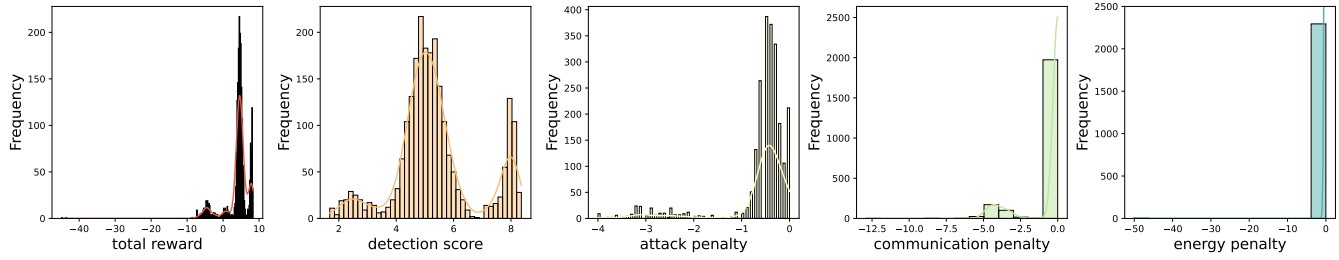

**Figure 10: Distributions of rewards and penalty terms on the ToN-Partition dataset**

## B.3   Specifications of platform and architecture of GNNs

MPGNN is implemented in PyTorch. The operation environment includes a 48-core 2.2GHz Intel Xeon Silver 4212 CPU, 64 GB of RAM, and the 64-bit Ubuntu 20.04 LTS operating system. Regarding software versions, Python 3.8, PyTorch 1.12, and CUDA 11.4 are used.

Regarding the implementation of the model, Actor is set as a two-layer GCN, with the first layer employing the ReLU activation function, while the output of the second layer is processed by the sigmoid function. And the number of hidden units is set as 128. Critic shares a similar structure with Actor, with the exception that its second layer produced a one-dimensional output without undergoing any activation function.

**Algorithm 3** The learning process of coalitional NIDSs

**Input:**

  Overlapped coalitions $\{C_1, C_2, \ldots, C_K\}$; States of IoT devices $\mathbf{X}$; Learning rate of E-GraphSAGE $LR_N$; Coalition head selecting function $\mathcal{R}$; The number of E-GraphSAGE layer $L$.

**Output:** Detection results for the entire IoT network $\hat{Y}$

1: **for** $\forall k = 1, 2, \ldots, K$ **do**
2:   Determine the coalition head: $h_k \leftarrow \max_i \mathcal{R}(X_i)$ s.t. $i \in C_k$
3:   Construct the attributed multigraph of $C_k$: $G_{\text{NID}_k} = (\mathcal{V}_{ip} \times \mathcal{V}_{port}, \mathcal{E}, \mathbf{E})$
     // Establish coalitional NIDS
4:   **for** $l \leftarrow 1$ **to** $L$ **do**
5:     **for** $\forall v \in \mathcal{V}$ **do**
6:       $\mathbf{h}^l_{\mathcal{N}(v)} \leftarrow \text{AGG}^l \left( \{ \mathbf{e}^{l-1}_{uv}, \forall u \in \mathcal{N}(v), uv \in \mathcal{E} \} \right)$
7:       $\mathbf{h}^l_v = \sigma \left( \mathbf{W}^l \text{concat} \left( \mathbf{h}^{l-1}_v, \mathbf{h}^l_{\mathcal{N}(v)} \right) \right)$
8:     **end for**
9:   **end for**
10:   $\mathbf{z}_v = \mathbf{h}^L_v$, $\mathbf{z}_{uv} = \text{concat}(\mathbf{z}_u, \mathbf{z}_v)$, $\hat{Y}_k = \text{MLP}(\mathbf{z}_{uv})$
11:   $\ell_{\text{NIDS}} = H(\hat{Y}_k, Y_k)$
12:   $\text{Detection}(G_{\text{NID}_k}; \hat{\theta}_k) \leftarrow \text{ADAM}(\theta_k, LR_N)$
      // Intrusion detection in each coalition
13:   Collect real-time network flows and construct attributed multigraphs: $G_{test_k} \leftarrow X_{test_k}$
14:   Identify the real-time network flows: $\hat{Y}_k \leftarrow \text{Detection}(G_{test_k}; \hat{\theta}_k)$
15: **end for**
16: Realize real-time intrusion detection of global network via ensemble prediction: $\hat{Y} \leftarrow \{\hat{Y}_1, \hat{Y}_2, \ldots, \hat{Y}_K\}$
17: **return** Detection results for the entire IoT network $\hat{Y}$

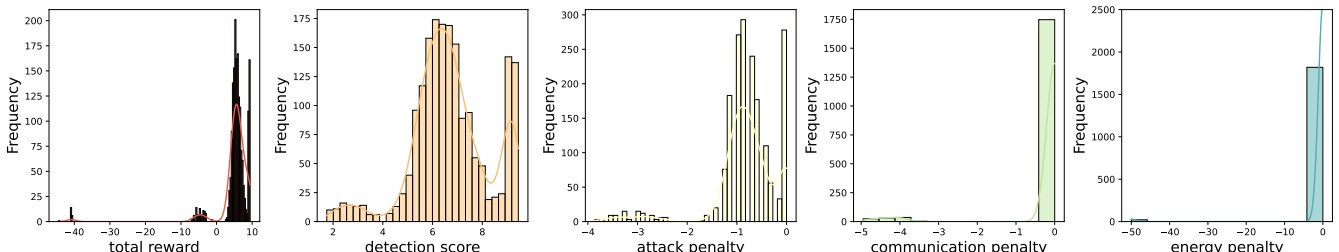

**Figure 11: Distributions of rewards and penalty terms on the CSE-Partition dataset**

Besides, two Adam optimizers with different learning rates are used to update Actor and Critic, respectively. As for the hyperparameters, the detailed settings are displayed in Table 7.

## B.4 Definition of Evaluation Metrics

Four widely used evaluation metrics for detection performance, namely accuracy (ACC), recall (REC), precision (PRE), and F1-score (F1), have been used in our evaluation. The accuracy quantifies the ratio of correctly classified samples among all samples, which can be calculated as follows:

$$ACC = \frac{TP + TN}{TP + FP + TN + FN} \tag{8}$$

The recall measures the proportion of all samples accurately identified attack samples to the total number of true attack samples, which can be calculated as follows:

$$REC = \frac{TP}{TP + FN} \tag{9}$$

The precision is defined as the proportion of accurately predicted attacks to the total number of samples predicted as attacks. The calculation can be expressed as:

$$PRE = \frac{TP}{TP + FP} \tag{10}$$

Table 7: The setting of hyperparameters

| Hyperparameter | Description | Default Value |
|---|---|---|
| $LR_\pi$ | The initial learning rate of the optimizer for Actor. | 5e-7 |
| $LR_Q$ | The initial learning rate of the optimizer for Critic. | 5e-4 |
| $LR_\alpha$ | The initial learning rate of the optimizer for $\alpha$. | 5e-5 |
| $\gamma$ | The future discount rate of rewards. | 0.9 |
| $\beta$ | The weight to balance the critic loss term $\mathcal{E}_\mathcal{D}$ and $\mathcal{L}_\mathcal{D}$. | 4 |
| $\tau$ | The update weight of the evaluation network to the target network. | 5e-3 |
| $\alpha_0$ | The initial value of $\alpha$. | 1 |
| N | The max training epochs for the offline reinforcement learning model. | 2000 |
| BATCH_SIZE | The number of training examples used in each iteration during the partition model training. | 5 |
| INTERATIONS | The number of gradient updates per epoch. | 2000 |
| REWARD_SCALE | The coefficient used to scale reward. | 1 |

And the F1-score is mathematically defined as the harmonic mean of precision and recall :

$$F1 = 2 \times \frac{Recall \times Precision}{Recall + Precision} \tag{11}$$

Here, the $TP, TN, FP, FN$ represent the numbers of true positives, true negatives, false positives, and false negatives, respectively.

In addition, three supplementary metrics, namely Training Time (TT), Memory Size (SIZE), and Average of Node Number (AN), are employed to evaluate the effectiveness of resource consumption reduction and scale reduction. Here, the training time refers to the duration required for attributed graph construction and intrusion detection model training, which accounts for the majority of the time in the process of detection model training. The memory size denotes the average of the memory usage of the constructed attributed graph in each coalition. Additionally, the average number of nodes measures the scale of the partitioned coalitions.

## C APPENDIX: SUPPLEMENTARY EXPERIMENTAL RESULTS

### C.1 Distribution of attack types in formed coalitions (NF-CSE-CIC-IDS2018-V2)

Fig.12 offers a visualization of attack types across all coalitions generated by Random, Louvain, and MPGNN on the NF-CSE-CIC-IDS2018-V2 dataset. As displayed in Fig. 12, it becomes evident that the data distribution of coalitions generated by Random and Louvain exhibits an evident imbalance, causing the exclusion of certain types of attacks. This imbalance impairs the detection performance. In contrast, coalitions derived from the MPGNN exhibit a more balanced data distribution, which stands to enhance the effectiveness of coalitional NIDSs.

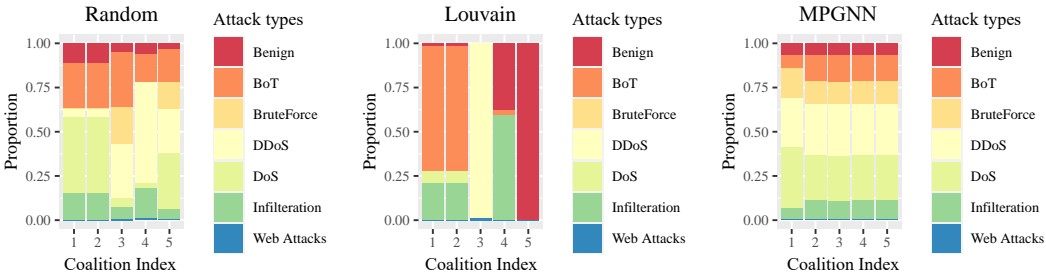

Figure 12: Distribution of attack types in formed coalitions (NF-CSE-CIC-IDS2018-V2)

### C.2 Normalized confusion matrix on NF-CSE-CIC-IDS2018-V2

Fig. 13 provides the visualization of normalized confusion matrices of centralized NIDS and MPGNN. As illustrated in Fig. 13, the detection capability of the coalitional NIDSs generated by MPGNN closely approximates that of centralized NIDS. Specifically, the recall rates for Bot, Brute-Force, and DoS reach a perfect ratio of 1.0. In contrast, the identification capability of Infiltration and Web attacks is relatively poor, similar to that of centralized NIDS. This phenomenon can be attributed to the constraints imposed by the limited sample size and the less distinct characteristics associated with these specific attack categories.

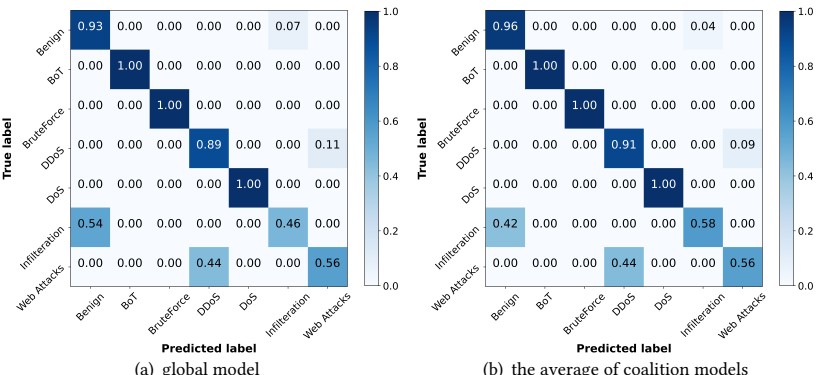

(a) global model      (b) the average of coalition models

**Figure 13: Normalized confusion matrices on NF-CSE-CIC-IDS2018-V2**

## C.3 Device capacities within Coalitions (NF-ToN-IoT-V2 Dataset)

Fig. 14 presents the computing resource of IoT nodes within coalitions generated by MPGNN. As depicted in Fig.14, the computing resource of IoT devices across coalitions exhibits a relatively balanced distribution, encompassing nodes with varying resource levels. Furthermore, it can be seen that each coalition has at least one high-end device that is capable of serving as the coalition head.

Coalition 1     Coalition 2     Coalition 3     Coalition 4     Coalition 5

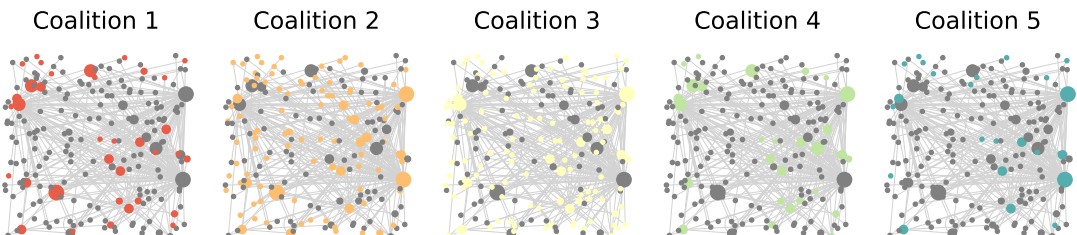

**Figure 14: Visualization of device capacities in formed coalitions (NF-ToN-IoT-V2)**

## C.4 Ablation experiments with different $\beta$ values

To assess the impact of $\beta$, which balances the weight between $\mathcal{L}_\mathcal{D}$ and $\mathcal{E}_\mathcal{D}$ in $\ell_Q$ as mentioned in Eqn. (3), we conduct an additional experiment. Specifically, we consider the values of $\{4^0, 4^1, 4^2, 4^3\}$, and corresponding results on the two datasets are detailed in Table 8.

**Table 8: Performance of MPGNN with different $\beta$**

| Dataset | Value | ACC | REC | PRE | F1 | TT(s) | SIZE(MB) | AN |
|---------|-------|-----|-----|-----|----|----|----------|-----|
| NF-ToN -IoT-V2 | $\beta = 1$ | 58.41% | 0.5949 | 0.5419 | 0.5353 | 285.03 | 100.38 | 112.6 |
| | $\beta = 4$ | 83.39% | 0.8581 | 0.8014 | 0.8187 | 450.86 | 153.75 | 61.0 |
| | $\beta = 16$ | 77.19% | 0.7730 | 0.7303 | 0.7270 | 460.54 | 159.64 | 162.2 |
| | $\beta = 64$ | 73.30% | 0.7574 | 0.7200 | 0.7052 | 399.07 | 147.09 | 93.6 |
| NF-CSE-CIC -IDS2018-V2 | $\beta = 1$ | 78.87% | 0.6995 | 0.6351 | 0.6141 | 393.80 | 191.19 | 57.2 |
| | $\beta = 4$ | 90.28% | 0.8337 | 0.8039 | 0.7769 | 454.76 | 229.46 | 62.2 |
| | $\beta = 16$ | 81.89% | 0.7529 | 0.7139 | 0.6943 | 385.03 | 204.16 | 65.8 |
| | $\beta = 64$ | 90.34% | 0.8462 | 0.8109 | 0.7836 | 564.41 | 250.85 | 158.2 |

## C.5 Ablation experiments regarding the reward functions

As detailed in Appendix A.1, the reward function of OGRL is defined as $r = r_{\text{NID}} - w_a \cdot p_a - w_c \cdot p_c - w_b \cdot p_b$, where $w_b$ is held constant at 50 to ensure the presence of at least one high-end node. In this subsection, we delve into the assessment of the effectiveness of the attack penalty $p_a$ and communication penalty $p_c$ by varying the values of $w_a$ and $w_c$. Specifically, we evaluate the coalitional NIDS performance

with distinct combinations of weights such as $\{(w_a = 1, w_c = 5),(w_a = 1, w_c = 1),(w_a = 5, w_c = 1)\}$, and provide corresponding results on both datasets in Table 9.

As outlined in Table 9, when $w_a$ is held constant and $w_c$ is reduced from 5 to 1, the results from both datasets reveal an increase in average node numbers, suggesting that the communication penalty plays a non-negligible role in controlling the scale of coalitions. Furthermore, we maintain $w_c$ at 1 while varying the value of $w_a$ from 1 to 5. We can see a considerable increase in memory consumption, and the scale of coalitions on both datasets. This phenomenon can be attributed to the rise in $w_a/w_c$, which emphasizes the role of the attack penalty in ensuring the inclusion of all attack categories within each coalition, while restricting the capacity of the communication penalty.

**Table 9: Performance of MPGNN with different reward function configurations**

| Dataset | Setting | ACC | REC | PRE | F1 | TT(s) | SIZE(MB) | AN |
|---|---|---|---|---|---|---|---|---|
| NF-ToN - IoT-V2 | $w_a = 1, w_c = 5$ | 83.39% | 0.8581 | 0.8014 | 0.8187 | 450.86 | 153.75 | 61.0 |
| | $w_a = 1, w_c = 1$ | 82.80% | 0.8545 | 0.8018 | 0.8128 | 453.39 | 164.08 | 123.2 |
| | $w_a = 5, w_c = 1$ | 79.58% | 0.8351 | 0.8008 | 0.7918 | 462.67 | 167.29 | 154.0 |
| NF-CSE-CIC -IDS2018-V2 | $w_a = 1, w_c = 5$ | 90.28% | 0.8337 | 0.8039 | 0.7769 | 454.76 | 229.46 | 62.2 |
| | $w_a = 1, w_c = 1$ | 79.56% | 0.7532 | 0.7147 | 0.6933 | 409.20 | 218.27 | 68.0 |
| | $w_a = 5, w_c = 1$ | 90.60% | 0.8531 | 0.8544 | 0.8123 | 464.86 | 248.46 | 76.2 |

