# OpenReview forum: "Divide, Conquer, and Coalesce: Meta Parallel Graph Neural Network for IoT Intrusion Detection at Scale"
_ACM.org/TheWebConf/2024/Conference — TheWebConf24_

### Official Review · Reviewer_jemh · 2023-11-08

**Novelty:** 4
**Technical Quality:** 5

**Review:**

### Summary
This paper proposes a meta parallel graph neural network (MPGNN) to serve as a scalable network intrusion detection system (NIDS). To improve the efficiency of NIDS while maintaining detection performance, MPGNN first employs a graph-embedded adversarially trained actor-critic (G-ATAC) algorithm to learn a good coalition formation (network partition) policy for the IoT network. Then MPGNN applies coalition-wise E-GraphSAGE to detect network intrusions in each coalition, formulated as an edge classification task. In experiments, MPGNN can outperform competitive baselines and achieve comparable performance to the centralized NIDS.
### Pros
- P1. The efficiency issue of NIDS is significant and well-motivated.
- P2. The paper is clearly written and well organized.
- P3. The technical quality of this paper is fairly good, except for the C2 and C3 listed below.
### Cons
- C1. Some important related research works are missing. The novelty is thereby impacted.
- C2. The proposed G-ATAC method lacks efficiency analysis.
- C3. Some claimed techniques (meta-learning) or experiments (DL-based partitioning) are missing.

**Questions:**

- Q1. Some important related research works are missing. There already exist many "federated network intrusion detection" methods, such as [1] and [2], and as listed in the survey [3]. There are even more if considering "federated network anomaly detection" methods. The authors might need to include some of them as related work.
- Q2. Due to the issue mentioned in Q1, state-of-the-art NIDS methods are not included in the experiments. The authors might need to include some of them as baselines.
- Q3. The G-ATAC coalition formation method operates on the entire IoT network. Would there be any efficiency issue for G-ATAC, especially if the IoT network is very large? There should be complexity analysis or running time comparisons for different coalition formation methods.
- Q4. Since G-ATAC is conducted offline using only the historical data, MPGNN performance could degrade in the future when the IoT network evolves with structural or feature changes.
- Q5. For the coalition formation method, how about network partitioning based on regions? The authors may consider this if there is region-related information in the dataset.
- Q6. Where is the DL-based partitioning mentioned in line 647? There seem to be no experimental results of this method in any table or figure.
- Q7. Where is the meta-learning mentioned in the abstract and the conclusion? There seems to be no meta-learning in the proposed methodology.
- Q8. Page 2 line 330, "conditional" should be corrected to "coalitional"

[1] Beibei Li, Yuhao Wu, Jiarui Song, Rongxing Lu, Tao Li, Liang Zhao: DeepFed: Federated Deep Learning for Intrusion Detection in Industrial Cyber-Physical Systems. IEEE Trans. Ind. Informatics 17(8): 5615-5624 (2021)

[2] Sawsan Abdul Rahman, Hanine Tout, Chamseddine Talhi, Azzam Mourad: Internet of Things intrusion Detection: Centralized, On-Device, or Federated Learning? IEEE Netw. 34(6): 310-317 (2020)

[3] Shaashwat Agrawal, Sagnik Sarkar, Ons Aouedi, Gokul Yenduri, Kandaraj Piamrat, Mamoun Alazab, Sweta Bhattacharya, Praveen Kumar Reddy Maddikunta, Thippa Reddy Gadekallu: Federated Learning for intrusion detection system: Concepts, challenges and future directions. Comput. Commun. 195: 346-361 (2022)

**Reviewer Confidence:**

3: The reviewer is confident but not certain that the evaluation is correct

**Scope:**

4: The work is relevant to the Web and to the track, and is of broad interest to the community

---

### Official Review · Reviewer_jZ2Y · 2023-11-11

**Novelty:** 4
**Technical Quality:** 4

**Review:**

Dear authors, thanks for submitting your work to WWW 2024. The paper has a great title + motivation, I do agree that partitioning the topology for modularity and scalability is a good idea, especially when it comes to IoT based intrusion detection with GNN-based models. The paper is well written overall and the evaluation section looks appealing. Figures are made informative and insightful. With all that being said, I do have major concerns about the proposed solution as well as evaluation methodology, please see below for details.

1. How does coalition formation affect coalitional NIDS: Impact on timeliness. This is a great research question by itself, and I do agree that both computational complexity and accuracy needs to be taken into account. I'm glad you mentioned networking could become the bottleneck due to all kind of reasons such as limited throughput and link failures, and I was looking forward to your optimizations and discussions on this angle. However, this argument was not mentioned in later sections of the paper. The evaluation section briefly mentions training time and memory size, but that doesn't really show the networking aspect. Yes, since you have much smaller graphs to deal with, it makes sense that the training is more efficient, but when I look at your figure 8 , I have deep concerns on whether your method really makes sense w.r.t. networking overhead. The concrete suggestion is to evaluate e.g. the average distance among nodes within each coalition. In other words, I believe the current evaluation strategy is insufficient to conclude the motivation.
2. How does coalition formation affect coalitional NIDS: Impact on accuracy. It is expected that a random coalition strategy performs poorly, it is also understandable things like Louvian methods do not work well due to neglected connections across subgraphs. But that does not justify why RL is the needed solution here. In my opinion, all the evaluated baselines are simply not strong enough. Consider the follow strawman solution that is very easy to implement: from each of your "high end devices", we do a weighted BFS to get a "blast radius" that should be managed and controlled by them. The blast radius could have overlapping with one another so that we do not miss things on the border. All nodes within a blast radius is considered a coalition. Doing this has two advantages: (1) networking-vise, we make sure that IoT devices could send their data to closer high end devices, saving latency overhead as well as failure possibility. (2) accuracy-wise, since resources in each blast radius are within limited hop with one another, they by nature has more correlation thus leading to better prediction accuracy. This is just one possible solution among many that one can think of, but the point is, baselines that are even weaker than random doesn't show the contribution of this paper. The concrete suggestion is to come with stronger baselines.
3. Technical contribution. It appears to me that the primary technical novelty of the paper is applying GNN to ATAC, but I'm not sure whether that is strong enough. The reward function rely on the result of yet another ML model, which leads to concerns on efficiency and accuracy.How long does the offline RL take? The concrete suggestion is to focus on the high level contribution of your RL techniques, before  talking about lower level details

**Questions:**

Why does your solution provider ideal timeliness and accuracy? There are many simpler heuristics that might also work, as I mentioned in the review. Why do you believe that RL based approach is the most capable solution? Moreover, why is the proposed RL method novel? How long does the offline RL take?

**Reviewer Confidence:**

3: The reviewer is confident but not certain that the evaluation is correct

**Scope:**

3: The work is somewhat relevant to the Web and to the track, and is of narrow interest to a sub-community

---

### Official Review · Reviewer_oMoH · 2023-11-23

**Novelty:** 5
**Technical Quality:** 5

**Review:**

The paper under review presents the Meta Parallel Graph Neural Network (MPGNN) for establishing a scalable Network Intrusion Detection System (NIDS) in large-scale Internet of Things (IoT) networks. The MPGNN framework leverages a meta-learning approach to optimize parallelism in GNN-based NIDS. A key feature of MPGNN is its coalition formation policy, which partitions a massive graph into multiple coalitions or subgraphs to maximize performance and efficiency. It introduces an offline reinforcement learning algorithm, Graph-Embedded Adversarially Trained Actor-Critic (G-ATAC), to learn this coalition formation policy, focusing on optimizing intrusion detection accuracy, communication overheads, and computational complexities. MPGNN employs E-GraphSAGE for establishing coalitional NIDSs, which then collaborate through ensemble prediction for network-wide intrusion detection. The paper reports substantial improvements in F1 scores and reductions in training time compared to centralized NIDS, based on evaluations using two real-world datasets​​.

Strengths:
1. MPGNN's approach to scaling NIDS for large IoT networks using a meta-learning framework and coalition formation policy is relatively innovative. It addresses the challenges of timeliness and computational complexity in GNN-based NIDS, critical for practical implementation in real-world IoT networks​​.
2. The use of an offline reinforcement learning algorithm (G-ATAC) for learning coalition formation is a robust methodological choice. It ensures the model learns from temporal dependencies of network states and decisions without the need for expensive online interactions​​.
3. The paper includes thorough experimental validation, demonstrating MPGNN's effectiveness over large datasets and providing empirical evidence of its superiority over state-of-the-art methods in terms of F1 scores and training efficiency​​.

Weaknesses:
1. G-ATAC is a sophisticated algorithm that may present significant challenges in terms of tuning and optimization. The complexity of this approach could lead to overfitting or difficulties in adapting the model to different IoT network environments. Is this something the proposed framework can mitigate?
2. In scenarios where the offline data is not sufficiently diverse or fails to capture the evolving nature of network threats, the MPGNN model's performance and adaptability could be significantly compromised due to the reliance on meta-learning for optimizing parallelism.
3. The evaluation of MPGNN focuses on two real-world datasets, but it lacks a detailed exploration of how the model performs across a diverse range of IoT scenarios, particularly those with unique network topologies or where real-time intrusion detection is critical.
4. The paper describes the implementation of coalitional NIDS using E-GraphSAGE for malicious flow detection. However, there is a lack of in-depth analysis of how effectively E-GraphSAGE handles the variations in network traffic and attack patterns within each coalition.
5. A significant challenge for GNN-based NIDS, as acknowledged in the paper, is the processing of massive graphs in large-scale IoT networks. The size of graph data can reach gigabytes, leading to substantial communication and latency overheads during graph construction. The computational complexity involved in processing these large graphs may exceed the capabilities of resource-constrained IoT devices, causing unacceptable detection delays. While MPGNN aims to address these issues, the paper does not provide a comprehensive solution or evaluation of its effectiveness in mitigating these challenges, particularly in extremely large or rapidly expanding IoT networks​​.
6. Missing related work comparison: Without a clear comparison with existing work, it's challenging for readers to understand how MPGNN fits into the broader landscape of NIDS research, particularly in the context of large-scale IoT networks. This makes it difficult to gauge the true novelty and significance of the proposed approach.
7. The evaluation relies on two datasets - the ToN-Partition and the CSE-Partition datasets. While these datasets provide a basis for testing, the size and diversity of the datasets (2296 and 1840 samples, respectively) may not fully represent the vast and varied nature of IoT networks. A larger and more diverse set of datasets would be more convincing in demonstrating the scalability and adaptability of MPGNN to different IoT environments.

In conclusion, the paper presenting the Meta Parallel Graph Neural Network (MPGNN) for Network Intrusion Detection Systems (NIDS) in IoT networks makes notable contributions to the field, particularly in addressing scalability and efficiency challenges in large-scale IoT environments. MPGNN's use of a meta-learning framework and coalition formation policy, coupled with the innovative Graph-Embedded Adversarially Trained Actor-Critic (G-ATAC) algorithm, demonstrates a unique approach to optimizing NIDS.

However, there are several areas where the paper could be strengthened:

1. Related Work: The absence of a dedicated section for related work limits the ability to contextualize MPGNN within the existing body of research and assess its novelty and advancements over current methodologies.
2. Evaluation Depth: While the evaluation employs relevant metrics and datasets, it is limited by the size and diversity of the datasets used and the range of attack types tested. This restricts the ability to fully assess the model's scalability and adaptability to different IoT environments and attack scenarios.
3. Practical Implementation Considerations: The paper could benefit from a more thorough discussion on practical aspects of deploying MPGNN in real-world IoT networks, considering dynamic network topologies, varying device capabilities, and real-time processing requirements.
4. Computational Efficiency and Scalability: Further details regarding the computational efficiency, particularly in extremely large-scale IoT networks, would strengthen the understanding of MPGNN's practical applicability.

The paper presents a step forward in the development of scalable and efficient NIDS for IoT networks. However, addressing these identified weaknesses could enhance its contribution to the field and provide a more comprehensive understanding of its applicability and performance in real-world IoT network scenarios.

**Questions:**

1. Can you provide additional insights into the choice of the ToN-Partition and CSE-Partition datasets? Are there plans to test MPGNN on larger and more diverse datasets to better assess its scalability and effectiveness across a broader range of IoT environments and attack types?
2. How does MPGNN adapt to dynamic IoT environments with varying network topologies and device capabilities? Could you provide more details on MPGNN's practical deployment considerations in real-world IoT settings?

**Reviewer Confidence:**

3: The reviewer is confident but not certain that the evaluation is correct

**Scope:**

3: The work is somewhat relevant to the Web and to the track, and is of narrow interest to a sub-community

---

### Official Review · Reviewer_BCyh · 2023-11-23

**Novelty:** 4
**Technical Quality:** 5

**Review:**

This paper introduces the Meta Parallel Graph Neural Network (MPGNN) to improve network intrusion detection in large-scale IoT environments. Traditional methods falter due to the complexity of modern networks. MPGNN utilizes a meta-learning framework to enhance the Graph Neural Network (GNN)-based Network Intrusion Detection Systems (NIDS) through optimized parallelism. It partitions vast network data into smaller, manageable coalitions, balancing detection accuracy, communication load, and computational demands. An offline reinforcement learning algorithm, Graph-Embedded Adversarially Trained Actor-Critic, develops policies for effective coalition formation. These policies handle network state temporal dependencies without expensive online processes. MPGNN incorporates E-GraphSAGE for in-coalition detection, with coalitions uniting via ensemble prediction for overall network security. Evaluations reveal MPGNN's superior detection accuracy and faster training, proving its effectiveness in scalable, efficient intrusion detection for extensive IoT networks.

Strengths:

++ Novelty: The authors introduce a novel solution to overcome the limitations of conventional centralized intrusion detection systems. By partitioning the network into distinct coalitions, MPGNN effectively addresses issues related to computational complexity, communication overhead, and detection accuracy.

++ Performance Improvements: MPGNN demonstrates notable improvements in various evaluation metrics, showcasing its superiority over existing methods in terms of efficiency and effectiveness.

++ Comprehensive Ablation Studies: The paper is commendable for conducting thorough ablation studies, providing robust evidence for the effectiveness and reliability of MPGNN's design.



Weaknesses:
-- The methodology for selecting coalition heads in MPGNN is inadequately explained. The criteria for selection, including computational capacity considerations, are not clearly outlined. Moreover, the use of basic diagrams to demonstrate computational capabilities is insufficient and lacks depth.

-- MPGNN's requirement for retraining with each new dataset limits its practicality in real-world applications. This lack of adaptability is a significant drawback for a model intended for diverse and dynamic datasets.

-- The paper's explanation of the reinforcement learning algorithm lacks detail. Based on the information provided in the paper, the requirement for the agent to assign each node to a specific coalition at every step results in an oversized action space. This complexity poses significant challenges to the training process. Additionally, a more detailed explanation, particularly through an example illustrating a single step in the reinforcement learning training sequence would be helpful. Such an addition would clarify its impact on the model's efficiency and its feasibility in practical scenarios.

-- The paper's writing is difficult to follow, lacking a coherent structure necessary for understanding the proposed method.

**Questions:**

Please refer to the strengths and weaknesses.

**Reviewer Confidence:**

2: The reviewer is willing to defend the evaluation, but it is likely that the reviewer did not understand parts of the paper

**Scope:**

3: The work is somewhat relevant to the Web and to the track, and is of narrow interest to a sub-community

---

### Official Review · Reviewer_XPWB · 2023-11-29

**Novelty:** 3
**Technical Quality:** 4

**Review:**

The paper was an interesting read, was well-written and tackles an intriguing topic. Overall, I liked to idea to intelligently create different coalitions, and divide the intrusion detection problem among them, resulting in improved performance in terms of time and memory. Nevertheless, I do have some comments:

**Practical benefits of improve performance**
The main (achieved) goal of the paper is to improve the performance of NIDS. It is not clear what the practical benefit of the achieved  41.63% (22.11%) reduction in training time is. Does this allow for new applications of NIDS that were previously not possible? It would be great if the paper would expand on these implications. Furthermore, there's only a comparison made against a centralized NIDS - it would be interesting to see how other NIDS approaches perform on the same datasets as well.

**Datasets**
The analysis is made on two datasets: NF-ToN-IoT-V2 and NF-CSE-CIC-IDS2018-V2. The former has a much higher ratio of attack traffic. The researchers then sample the data according to the respective attack distributions of both datasets. If I understand that correctly (please add the information of the datasets after pre-processing), the final dataset has the same ratio of benign/malicious samples. This seems like a rather unrealistic ratio, so I am curious how this affects the performance of the various NIDS approaches: e.g. random coalition formation seems to be significantly better for NF-ToN-IoT-V2 than NF-CSE-CIC-IDS2018-V2. Furthermore, I am missing an intuition in how the size of the dataset would affect the results (e.g. how would the performance be if the dataset was 2x, 4x, ... larger?)
Finally, ... has been shown to have certain flaws and an improved version is available [1].

**

[1]: Error Prevalence in NIDS datasets: A Case Study on CIC-IDS-2017 and CSE-CIC-IDS-2018 - Liu et al. (CNS'22)

**Questions:**

Why was the number of coalitions fixed to 5?

**Ethics Review Description:**

No ethics concerns

**Reviewer Confidence:**

2: The reviewer is willing to defend the evaluation, but it is likely that the reviewer did not understand parts of the paper

**Scope:**

3: The work is somewhat relevant to the Web and to the track, and is of narrow interest to a sub-community

---

### Decision · Program_Chairs · 2024-01-22

**Decision:**

Accept

**Comment:**

The paper enjoys both an above average level of novelty as well as technical contributions. The major requests were to better describe the methodology, provide a better comparison with related work, and clarify the overhead for re-training.
 The authors provided extensive replies to the comments raised by reviewers. This led two reviewers to raise their score, under the promise from the authors to accommodate in the final version f the paper (should it be accepted) the highlighted modifications.
 With the promised modifications implemented, the paper would definitely rank in the "Accept" batch.

 ---